# $\mathcal{I}/c$-Extremization in M/F-Duality

Marieke van Beest, Sebastjan Cizel, Sakura Schäfer-Nameki, James Sparks

*Mathematical Institute, University of Oxford*

*Woodstock Road, Oxford, OX2 6GG, United Kindom*

We study the holographic dual to $c$-extremization for 2d $(0,2)$ superconformal field theories (SCFTs) that have an AdS$_3$ dual realized in Type IIB with varying axio-dilaton, i.e. F-theory. M/F-duality implies that such AdS$_3$ solutions can be mapped to AdS$_2$ solutions in M-theory, which are holographically dual to superconformal quantum mechanics (SCQM), obtained by dimensional reduction of the 2d SCFTs. We analyze the corresponding map between holographic $c$-extremization in F-theory and $\mathcal{I}$-extremization in M-theory, where in general the latter receives corrections relative to the F-theory result.

## 1 Introduction

A Type IIB supergravity solution with a holomorphically varying axio-dilaton can be given an F-theory interpretation, whereby the varying axio-dilaton is represented as the complex

structure of a singular elliptic fibration over spacetime. The axio-dilaton undergoes $SL(2,\mathbb{Z})$ monodromy around the singularities, which in turn encode the 7-branes in the background. Although such F-theory backgrounds have mainly been studied in the context of Minkowski solutions, recently AdS solutions were developed that have the hallmark of a holomorphically varying axio-dilaton. A class of AdS$_3$ solutions of F-theory were obtained in [1, 2], generalizing the constant $\tau$ solutions in e.g. [3–12]. The dual field theories are obtained by wrapping D3-branes on curves, above which the axio-dilaton varies. From the point of view of the 4d $\mathcal{N}=4$ Super-Yang-Mills (SYM) theory on the D3-branes, this corresponds to a varying complexified coupling, and the 2d SCFT is obtained by a duality-twist [13–16]. Generalizations of these F-theory solutions were obtained in [17–20] and the dual field theories were studied in [21, 22].

The most concrete avenue for accessing the geometric interpretation of F-theory is through its duality with M-theory, taken in its low-energy limit as 11d supergravity. The advantage of the M-theory dual perspective is that the elliptic fibration associated to the varying axio-dilaton appears as part of the physical spacetime geometry, rather than as an auxiliary space. There are two ways of dualizing the F-theory solutions, which in terms of dual field theory realized on D3-branes correspond to either mapping D3-branes to M5-branes or to M2-branes, depending on whether the T-duality is applied transverse to or along the world-volume of the D3-branes, respectively. In the former case, the F-theory solutions of [1] map to AdS$_3$ solutions in M-theory, which are dual to the MSW-strings [23]. Alternatively, by writing the AdS$_3$ as a constant-sized circle fibration over AdS$_2$, one can dualize along the fiber and obtain an AdS$_2$ solution of M-theory.

Among these AdS$_3$ solutions, the least well-understood are dual to 2d $\mathcal{N}=(0,2)$ super-symmetry [2], where, unlike for $(0,4)$[1], a general classification is not known. More importantly, the central charge of the 2d SCFTs for $(0,2)$ supersymmetry have to be determined by an extremization principle. Likewise, the dual M-theory AdS$_2$ backgrounds are holographic duals to 1d $\mathcal{N}=(0,2)$ SCQMs living on the conformal boundary, whose 1d partition function also requires an extremization. Indeed, in a supersymmetric 2d $(0,2)$ field theory the R-symmetry $U(1)_R$ is in general not uniquely defined. If the theory has other global abelian symmetries, these may mix with the $U(1)_R$ to produce an equally good R-symmetry. On the other hand, if the field theory flows to a superconformal fixed point in the infrared, this singles out a unique superconformal R-symmetry. In [24] an extraordinarily simple method for determining the exact R-symmetry of the fixed point SCFT was obtained, starting from

---

[1]The most general solutions with varying axio-dilaton and five-form flux dual to 2d $(0,4)$ were determined in [1].

the gauge theory description. The authors of [24] showed that extremizing the central charge $c$ of the field theory over all admissible R-symmetries exactly identifies the superconformal R-symmetry. This so-called $c$-extremization was further developed for compactifications of D3-branes with constant and varying coupling in [2, 25, 26]. It is closely related to the 4d concept of $a$-maximization, which was formulated in [27]. A related concept, which will also play a central role in this paper, is $\mathcal{I}$-extremization. This has been proposed in [28] as a method for determining the exact R-symmetry of an SCQM, when this theory arises as a compactification of a 3d $\mathcal{N} = 2$ SCFT on a Riemann surface $\Sigma$. Then the topologically twisted index of the 3d theory is expected to yield the 1d partition function [29–31] after extremization.

The AdS/CFT correspondence implies that equivalent geometric extremization principles should exist, realizing the gravitational duals of $\mathcal{I}$- and $c$-extremization. Indeed such geometric duals were constructed in [32, 33] (as well as [34, 35] for $a$-maximization) for backgrounds where the axio-dilaton is constant. These papers formulate holographic extremization principles for the geometries in [5, 7, 36] by determining a parametrization of the Killing vector that is the geometric counterpart to the R-symmetry, and the conditions for the optimization problem to be well-defined, as well as the geometric quantity to be extremized. A general proof of the off-shell holographic correspondence with $\mathcal{I}$- and $c$-extremization was put forth in [37] and extended in [38]. A complementary approach using gauged supergravity was pursued in [39, 40].

Whether one takes the geometric or field theoretic point of view, the extremization principles generally represent a significant simplification of the problem of determining genuine supergravity backgrounds or, equivalently, their dual SCFTs. Instead of directly having to solve a set of coupled (partial) differential equations, we can take the much more technically tractable approach of optimizing a single function.

In this paper we generalize this approach to include a varying axio-dilaton, which provides a powerful tool for identifying F-theory AdS$_3$ supergravity solutions that can arise from configurations of D3-branes and 7-branes. Furthermore, from this point of view the duality with M-theory AdS$_2$ backgrounds not only provides a description where the elliptic fibration associated with the varying axio-dilaton is physically manifest, it also implies that in this specific context holographic $\mathcal{I}$- and $c$-extremization are dual to each other.

More precisely, we will find that in general the two quantities obtained by extremization in M/F-theory only agree up to leading order in an expansion in terms of the volume of the

elliptic fiber. Namely, we find

$$\log Z_{1d} = \frac{1}{4G_2} = \frac{\Delta\phi}{12} c_{\text{sugra}} + \mathcal{O}(k_0) = \sqrt{\frac{2}{3}} \pi N_0^{1/2} \cdot c_{\text{sugra}}^{1/2} + \mathcal{O}(k_0) \,, \qquad (1.1)$$

where $k_0$ is the volume of the elliptic fiber. Here on the left hand side $Z_{1d}$ is the partition function of the 1d SCQM, which via holography is related to the 2d Newton constant $G_2$ of the dual M-theory $AdS_2$ solution. On the right hand side $c_{\text{sugra}}$ is the leading order 2d central charge of the F-theory $AdS_3$ solution, $N_0 \in \mathbb{N}$ is a certain quantized flux number, while $\Delta\phi$ is the size of the circle upon which the 2d SCFT is compactified. The correction terms in (1.1) are $\mathcal{O}(k_0)$. In M-theory, this fiber volume is a physical quantity, whereas in F-theory the elliptic fiber is an auxiliary geometric structure, where only the complex structure has a physical meaning, and the volume is strictly taken to zero. The correction terms then generically arise because the M-theory backgrounds include the full backreaction of the 7-branes on the F-theory side, which in particular break the circle isometry on which we T-dualise.

Let us conclude by making some comments on the physical interpretation of (1.1). The left hand side is the logarithm of the 1d partition function of the SCQM on a circle. On the other hand, the first expression involving $c_{\text{sugra}}$ is precisely the *Casimir energy* of the 2d $(0,2)$ theory placed on a torus, as one might have expected on general grounds. The final expression on the right hand side of (1.1) is proportional to $c_{\text{sugra}}^{1/2}$, with a proportionality constant that is a fixed number. In particular, this shows that to leading order in $k_0$ the two extremization principles are dual to each other. We shall see explicit examples, where the $\mathcal{O}(k_0)$ correction terms are either zero or non-zero.

The paper is organized as follows: In section 2 we present the details of the supersymmetric F-theory $AdS_3$ geometries and generalize the method of holographic $c$-extremization to accommodate a varying axio-dilaton. In section 3 we review the M/F-duality for the supersymmetric $AdS_2/AdS_3$ geometries and specialize holographic $\mathcal{I}$-extremization to the case where the compactification space contains a non-trivial elliptic fibration. We then determine the map between $\mathcal{I}$- and $c$-extremization in section 4. In section 5 we consider a large class of toric examples and apply $\mathcal{I}/c$-extremization to a novel set of M/F-theory setups, and rederive a known class of solutions, the elliptic surface universal twist solutions, using this new framework. For these theories the M- and F-theory computations agree without any corrections. Finally, section 6 contains an analysis of a related known class of M/F-theory solutions, the elliptic three-fold universal twist solutions, where the M-theory result for $1/G_2$ receives corrections compared to the F-theory computation. We conclude in section 7.

## 2 Holographic $c$-Extremization in F-Theory

We develop the holographic dual to $c$-extremization in the context of $\mathrm{AdS}_3$ geometries in F-theory, i.e. Type IIB supergravity with a holomorphically varying axio-dilaton $\tau$, which are holographically dual to 2d $\mathcal{N} = (0,2)$ SCFTs. To begin with we review the class of geometries [2], before generalizing holographic $c$-extremization in Type IIB [32] to encompass these F-theory geometries.

### 2.1 $\mathrm{AdS}_3$ Backgrounds

We consider holographic duals to 2d $(0,2)$ SCFTs realized in Type IIB with five-form flux and varying axio-dilaton [2]. The geometry underlying the solutions is $\mathrm{AdS}_3 \times Y_7$, and is supported by RR five-form flux

$$
\begin{aligned}
\mathrm{d}s_{10}^2 &= L_{10}^2 \, \mathrm{e}^{-B_{10}/2} \left[ \mathrm{d}s^2(\mathrm{AdS}_3) + \mathrm{d}s^2(Y_7) \right] , \\
F_5 &= -L_{10}^4 \left( \mathrm{vol}_{\mathrm{AdS}_3} \wedge F + *_7 F \right) .
\end{aligned}
\tag{2.1}
$$

In addition the axio-dilaton varies over the space $Y_7$. Here $L_{10}$ is an overall length scale, and $B_{10}$ and $F$ are a function and a closed two-form on $Y_7$, respectively. The analysis of the supersymmetry equations reveals that $Y_7$ admits a nowhere vanishing Killing vector $\xi$, which is the geometric counterpart to the $U(1)$ R-symmetry of the dual $(0,2)$ SCFT. The Killing vector induces a transversely conformally Kähler foliation $\mathcal{F}_\xi$. This entails that there is a locally defined space transverse to $\xi$, which we will denote by $\mathcal{M}_6$, admitting a Kähler metric. The geometric picture is most straightforward for the case of a *quasi-regular* Killing vector. By definition this means that the orbits of $\xi$ close and $Y_7$ is the total space of the circle fibration

$$
\begin{array}{ccc}
S^1 & \hookrightarrow & Y_7 \\
& & \downarrow \\
& & \mathcal{M}_6
\end{array}
\tag{2.2}
$$

where the transverse Kähler space $\mathcal{M}_6$ is a compact Kähler orbifold. The R-symmetry is then globally a $U(1)$ symmetry. When the generic orbits do not close, the Killing vector is said to be *irregular*. For a more detailed description of the general properties of $\mathcal{F}_\xi$ we refer the reader to [32]. The brane configuration corresponding to these geometries consists of $N$ D3-branes on $\mathbb{R}^{1,1} \times C$, where $C$ are curves in $\mathcal{M}_6$, above which the axio-dilaton varies. The auxiliary elliptic fiber degenerates over the loci that are subspaces wrapped by the 7-branes, which in the present case have world-volume $\mathcal{W}_8 = \mathrm{AdS}_3 \times \widetilde{S}$, where $\widetilde{S}$ are five-cycles in $Y_7$.

The supersymmetry equations of Type IIB get modified when the axio-dilaton is varying. The $SL(2,\mathbb{Z})$ self-duality of Type IIB induces a so-called $U(1)_D$ symmetry, which acts on the fermions and supercharges by

$$U(1)_D: \qquad \gamma = \left(\begin{smallmatrix} a & b \\ c & d \end{smallmatrix}\right) \in SL(2,\mathbb{Z}): \qquad \mathrm{e}^{\mathrm{i}\alpha_\gamma} = \frac{|c\tau + d|}{c\tau + d}. \tag{2.3}$$

The action on the fermions with half-integral charge extends the $SL(2,\mathbb{Z})$ by a $\mathbb{Z}_2$ to the metaplectic group [41]. The duality $U(1)$-symmetry $U(1)_D$ can be gauged, and then defines a line bundle $\mathcal{L}$, with connection

$$Q = -\frac{1}{2\tau_2}\mathrm{d}\tau_1, \tag{2.4}$$

where $\tau = \tau_1 + \mathrm{i}\tau_2$. Furthermore, it is convenient to define the one-form

$$\mathcal{P} = \frac{\mathrm{i}}{2\tau_2}\mathrm{d}\tau. \tag{2.5}$$

Supersymmetry implies that $\tau$ is preserved by $\xi$ (i.e. $\mathcal{L}_\xi \tau = 0$) and that it varies holomorphically over the transverse Kähler space. The bundle $\mathcal{L}$ is then transversely holomorphic with the curvature given by

$$\mathrm{i}\,\mathrm{d}\mathcal{P} = \mathrm{d}Q = -\mathrm{i}\mathcal{P} \wedge \bar{\mathcal{P}}. \tag{2.6}$$

Next we consider how the geometry of $Y_7$ itself is constrained by supersymmetry. Let $\eta$ be the one-form dual to $\xi$. Choosing a local coordinate $z$ so that $\xi = 2\partial_z$, the local expression for $\eta$ is given by $\eta = \frac{1}{2}(\mathrm{d}z + P)$.[2] The derivative of the local one-form $P$ then satisfies

$$\mathrm{d}P = \rho_6 - \mathrm{i}\mathcal{P} \wedge \bar{\mathcal{P}}, \tag{2.7}$$

where $\rho_6$ is the transverse Ricci form. Finally, there is a relation between the scalar curvature $R_6$ of the transverse Kähler space and the warp factor $B_{10}$

$$\mathrm{e}^{B_{10}} = \frac{1}{8}\left(R_6 - 2|\mathcal{P}|^2\right). \tag{2.8}$$

Before proceeding let us summarize all the expressions for 10d fields after having imposed

---

[2]Note that we are following the conventions in [32], which are different from the conventions in [2]. The naming differences are particularly subtle when it comes to the connection one-forms. The reader should be aware that $P_{\mathrm{here}} = -\rho_{\mathrm{there}}$, $\mathcal{P}_{\mathrm{here}} = P_{\mathrm{there}}$ and $(\rho_6)_{\mathrm{here}} = (\mathfrak{R}_6)_{\mathrm{there}}$.

the supersymmetry equations:

$$ds_{10}^2 = L_{10}^2\, e^{-B_{10}/2} \left[ ds^2\,(\text{AdS}_3) + \frac{1}{4}\,(dz + P)^2 + e^{B_{10}}ds^2\,(\mathcal{M}_6) \right],$$

$$F_5 = -L_{10}^4 \left( \text{vol}_{\text{AdS}_3} \wedge F + *_7 F \right),$$

$$F = -2J_6 + \frac{1}{2}d\left[ e^{-B_{10}}\,(dz + P) \right], \tag{2.9}$$

$$dP = \rho_6 - i\mathcal{P} \wedge \bar{\mathcal{P}},$$

$$e^{B_{10}} = \frac{1}{8}\left( R_6 - 2|\mathcal{P}|^2 \right).$$

Here $J_6$ is the Kähler form on $\mathcal{M}_6$. Notice that all of the 10d fields are completely determined by the transverse Kähler metric together with the line bundle $\mathcal{L}$. We refer to $Y_7$ satisfying the supersymmetry equations, and therefore having all the properties outlined above, as a *supersymmetric geometry*. For constant axio-dilaton $\mathcal{P} = 0$, and the above reduce to the Type IIB equations in [5].

All of the above results hold off-shell, by which we mean that we merely impose supersymmetry, without imposing the equations of motion. For supersymmetric geometries the equations of motion reduce to a PDE on the transverse Kähler space involving the metric and the connection on the line bundle $\mathcal{L}$. This is referred to as the *master equation* in [2], and is given by

$$\Box_6(R_6 - 2|\mathcal{P}|^2) = \frac{1}{2}R_6^2 - (R_6)_{\mu\nu}(R_6)^{\mu\nu} + 2|\mathcal{P}|^2 R_6 - 4(R_6)_{\mu\nu}\mathcal{P}^\mu\bar{\mathcal{P}}^\nu. \tag{2.10}$$

Geometries satisfying this equation will be called *on-shell*, and are solutions of the Type IIB supergravity equations with varying axio-dilaton, provided that the five-form flux is appropriately quantized. We will return to the flux quantization conditions in a later section.

The F-theory perspective amounts to giving the varying axio-dilaton a geometric interpretation in terms of an auxiliary elliptic fibration

$$\begin{array}{c} \mathbb{E}_\tau \lhook\joinrel\longrightarrow \mathcal{M}_8^\tau \\ \downarrow \\ \mathcal{M}_6 \end{array}. \tag{2.11}$$

The total space $\mathcal{M}_8^\tau$ is Kähler but not Calabi-Yau[3]. Locally, away from the singular fibers, the metric on the total space is

$$ds^2(\mathcal{M}_8^\tau) = \frac{1}{\tau_2}\left[ (d\psi + \tau_1 d\phi)^2 + \tau_2^2 d\phi^2 \right] + ds^2(\mathcal{M}_6). \tag{2.12}$$

---

[3]We will denote spaces which enjoy an elliptic fibration with a superscript $\tau$, indicating the complex structure of the elliptic fiber.

The master equation can then be interpreted as a curvature condition on the total space $\mathcal{M}_8^\tau$. Taking this view, the master equation is

$$\Box_8(R_8) = \frac{1}{2}R_8^2 - (R_8)_{\mu\nu}(R_8)^{\mu\nu}\,, \tag{2.13}$$

which is precisely the form of the equation for constant axio-dilaton, just in two dimensions higher.

## 2.2 Supersymmetric Action

A geometric dual of $c$-extremization was recently developed in [32] for Type IIB AdS$_3$ geometries with 2d $(0, 2)$ duals and *constant* axio-dilaton. A key step in constructing the geometric extremization problem was deriving a certain geometric function called the *supersymmetric action*. Solutions to the master equation are extrema of this action, and the corresponding extremal value can be used to compute the central charge of the dual SCFT. In this section, we generalize this action to backgrounds with varying axio-dilaton.

The Type IIB supergravity equations including varying $\tau$ are [2]

$$R_{\mu\nu} = 2\mathcal{P}_{(\mu}\bar{\mathcal{P}}_{\nu)} + \frac{1}{96}(F_5)_{\mu\sigma_1\cdots\sigma_4}(F_5)_\nu{}^{\sigma_1\cdots\sigma_4}\,, \qquad \mathrm{d}*F_5 = 0\,, \tag{2.14}$$

where $\mu, \nu = 0, 1, ..., 9$. Writing out the components of the Einstein equations along the internal space $Y_7$ we obtain

$$
\begin{aligned}
0 = {}& R_{7ab} - 2\mathcal{P}_{(a}\bar{\mathcal{P}}_{b)} + \frac{1}{2}\nabla_a B_{10}\nabla_b B_{10} + 2\nabla_{ab}B_{10} + \frac{1}{4}\nabla^2 B_{10}g_{7ab} - \frac{1}{2}(\mathrm{d}B_{10})^2\, g_{7ab} \\
& + \frac{1}{2}\mathrm{e}^{2B_{10}}F_{ac}F_b{}^d - \frac{1}{8}\mathrm{e}^{2B_{10}}F^2 g_{7ab}\,,
\end{aligned}
\tag{2.15}
$$

where $a, b = 1, 2, ..., 7$. This arises by extremizing the following action functional

$$S_F = \int_{Y_7} \mathrm{e}^{-2B_{10}}\left[R_7 - 2|\mathcal{P}|^2 - 6 + \frac{9}{2}(\mathrm{d}B_{10})^2 + \frac{1}{4}\mathrm{e}^{2B_{10}}F^2\right]\mathrm{vol}_{Y_7}\,, \tag{2.16}$$

with respect to the 7d metric, and generalizes the action functional for constant $\tau$ in [7]. Varying the other fields in this action gives rise to the remaining Type IIB equations of motion.

We now specialize to the case where $Y_7$ is supersymmetric. Using the notation introduced in the previous subsection, the metric on $Y_7$ can be written as

$$\mathrm{d}s^2\,(Y_7) = \eta^2 + \mathrm{e}^{B_{10}}\mathrm{d}s^2\,(\mathcal{M}_6)\,, \tag{2.17}$$

where $\mathrm{d}s^2 \left( \mathcal{M}_6 \right)$ is the transverse Kähler metric. Writing out the Ricci scalar we obtain (up to total derivatives)

$$R_7 = \mathrm{e}^{-B_{10}} R_6 - 5\mathrm{e}^{-B_{10}} \left( \mathrm{d}B_{10} \right)^2 - \frac{1}{16} \mathrm{e}^{-2B_{10}} (\mathrm{d}P)^2 \,. \tag{2.18}$$

Furthermore, the flux term in the action is

$$\frac{1}{4} \mathrm{e}^{2B_{10}} F^2 = 6 - \frac{1}{2} \mathrm{e}^{-B_{10}} \left( R_6 - 2|\mathcal{P}|^2 \right) + \frac{1}{16} \mathrm{e}^{-2B_{10}} (\mathrm{d}P)^2 + \frac{1}{2} \mathrm{e}^{-B_{10}} \left( \mathrm{d}B_{10} \right)^2 \,. \tag{2.19}$$

Combining these, we find that the action evaluated on supersymmetric geometries is given by

$$\begin{aligned}
S_F &= \frac{1}{2} \int_{Y_7} \mathrm{e}^{-3B_{10}} \left( R_6 - 2|\mathcal{P}|^2 \right) \mathrm{vol}_{Y_7} \\
&= \int_{Y_7} \eta \wedge \left( \rho_6 - \mathrm{i}\mathcal{P} \wedge \bar{\mathcal{P}} \right) \wedge \frac{J_6^2}{2} \,.
\end{aligned} \tag{2.20}$$

We may rewrite this in a slightly nicer way as follows. Notice that $\mathrm{i}\mathcal{P} \wedge \bar{\mathcal{P}}$ is the curvature of the connection (2.4) and hence is a representative of $2\pi c_1(\mathcal{L})$. The action only depends on the cohomology class and not the particular representative, so we can rewrite it in terms of $c_1(\mathcal{L})$ as

$$S_F = \int_{Y_7} \eta \wedge \left( \rho_6 - 2\pi c_1(\mathcal{L}) \right) \wedge \frac{J_6^2}{2} \,. \tag{2.21}$$

For fixed R-symmetry vector $\xi$ this function depends only on the transverse Kähler class of the Kähler form $J_6$, and here also the first Chern class of the line bundle $\mathcal{L}$.

The central charge of the dual 2d (0,2) SCFT is computed from the Brown-Henneaux formula [42]

$$c_{\mathrm{sugra}} = \frac{3L_{10}}{2G_3} \,, \tag{2.22}$$

where

$$\frac{1}{G_3} = \frac{L_{10}^7}{G_{10}} \int_{Y_7} \mathrm{e}^{-2B_{10}} \mathrm{vol}_{Y_7} \tag{2.23}$$

is the effective 3d Newton constant, and $G_{10}$ is the 10d Newton constant. For an off-shell supersymmetric geometry the trial central charge is

$$c_{\mathrm{trial}} = \frac{12(2\pi)^2}{\nu_3^2} S_F \,, \tag{2.24}$$

where

$$\nu_3 = \frac{2(2\pi l_s)^4}{L_{10}^4} \,. \tag{2.25}$$

An F-theory solution necessarily extremizes $c_{\mathrm{trial}}$ over the class of off-shell geometries, and the central charge of the holographic dual SCFT is determined by

$$c_{\mathrm{sugra}} = \frac{12(2\pi)^2}{\nu_3^2} S_F \bigg|_{\mathrm{on\text{-}shell}} \,. \tag{2.26}$$

## 2.3 Flux Quantization

To have a genuine solution of string theory, the five-form flux must be appropriately quantized. We now describe how to quantize the flux for off-shell supersymmetric geometries, which is essential for completing the setup of the extremization problem.

The Type IIB flux quantization conditions are

$$\frac{1}{(2\pi l_s)^4} \int_{S_\alpha} F_5 = N_\alpha^F \in \mathbb{Z}\,, \tag{2.27}$$

with five-cycles $S_\alpha \in H_5(Y_7, \mathbb{Z})$, and the restriction of the five-form flux to $Y_7$ is given by [2]

$$F_5|_{Y_7} = \frac{L_{10}^4}{4} \left[ (\mathrm{d}z + P) \wedge \left( \rho_6 - \mathrm{i}\mathcal{P} \wedge \bar{\mathcal{P}} \right) \wedge J_6 + \frac{1}{2} *_6 \mathrm{d} \left( R_6 - 2|\mathcal{P}|^2 \right) \right]\,. \tag{2.28}$$

They can also be expressed as

$$\nu_3 N_\alpha^F = \int_{S_\alpha} \eta \wedge \left( \rho_6 - 2\pi c_1(\mathcal{L}) \right) \wedge J_6\,. \tag{2.29}$$

For the flux quantization conditions to be well-defined we have to ensure that the integrals in (2.27) do not depend on the representatives of the cycles $S_\alpha$. This is automatic for on-shell geometries since the master equation is equivalent to $F_5$ being closed. For off-shell geometries the setup requires some additional care.

To make the flux quantization conditions well-defined we will impose two topological assumptions. Firstly, we require that

$$H^2(Y_7, \mathbb{R}) \cong H_B^2(\mathcal{F}_\xi)/\left[\rho_6 - 2\pi c_1(\mathcal{L})\right]\,. \tag{2.30}$$

We have introduced the basic cohomology groups $H_B^*$ of the foliation $\mathcal{F}_\xi$ that are formed by restricting the exterior derivative to $\xi$-invariant differential forms. Note that $[\rho_6 - 2\pi c_1(\mathcal{L})]$ is a closed basic class that is exact in $H^2(Y_7, \mathbb{R})$ due to (2.7). This condition is most transparent in the quasi-regular case, where it implies that every cycle in $H_5(Y_7, \mathbb{Z})$ admits a representative which is a circle-bundle over a four-cycle in $\mathcal{M}_6$.

The first condition by itself is not sufficient for the flux quantization conditions to be well-defined. The issue resides in the fact that even if we choose representatives $S_\alpha$ and $S_\beta$ in the same homology class and both tangent to $\xi$, the flux integrals can still be different. To see this explicitly note that, by the first assumption, any two cycles that differ by the Poincaré dual (PD) cycle $\alpha[\rho_6 - 2\pi c_1(\mathcal{L})]_{\mathrm{PD}}$ are homologous. However, for such cycles

$$\int_{S_\alpha} \eta \wedge (\rho_6 - 2\pi c_1(\mathcal{L})) \wedge J_6 - \int_{S_\beta} \eta \wedge (\rho_6 - 2\pi c_1(\mathcal{L})) \wedge J_6 = \alpha \int_{Y_7} \eta \wedge (\rho_6 - 2\pi c_1(\mathcal{L}))^2 \wedge J_6\,. \tag{2.31}$$

Therefore the second condition we impose is the vanishing of the integral on the right side, i.e.

$$\int_{Y_7} \eta \wedge (\rho_6 - 2\pi c_1(\mathcal{L}))^2 \wedge J_6 = 0 \,. \tag{2.32}$$

Imposing these two conditions on the off-shell geometries ensures that the flux quantization is indeed well-defined [32]. Furthermore, (2.32) is just the integrated version of the master equation, which follows by writing the latter as

$$\square_6 \left( R_6 - 2|\mathcal{P}|^2 \right) = (J_6 \wedge J_6)_\lrcorner \left[ \left( \rho_6 - \mathrm{i}\mathcal{P} \wedge \bar{\mathcal{P}} \right) \wedge \left( \rho_6 - \mathrm{i}\mathcal{P} \wedge \bar{\mathcal{P}} \right) \right] \,. \tag{2.33}$$

Integrating this equation over $Y_7$, the left hand side vanishes using Stokes' theorem. Using the identity

$$[(J_6 \wedge J_6)_\lrcorner (a \wedge a)] \frac{J_6^3}{3!} = 2a^2 \wedge J_6 \,, \tag{2.34}$$

we obtain precisely (2.32).

## 2.4 The Complex Cone and the Geometric Extremization Problem

Having set up the abstract extremization problem, we now turn to the question of how to parametrize the class of off-shell supersymmetric geometries over which we extremize the action, by constructing a complex cone associated to $Y_7$ that allows us to parametrize the space of R-symmetry vectors on $Y_7$.

Consider the cone $C(Y_7)$ with metric

$$\mathrm{d}s^2(C(Y_7)) = \mathrm{d}r^2 + r^2 \mathrm{d}s^2(Y_7) \,, \tag{2.35}$$

where $r \in \mathbb{R}_{>0}$. As for the constant axio-dilaton case, we can consider the natural, locally defined $(4,0)$-form on the cone that is given by

$$\Omega_{(4,0)} = \mathrm{e}^{\mathrm{i}z} \mathrm{e}^{3B_{10}/2} r^3 \left[ \mathrm{d}r - \frac{\mathrm{i}r}{2} \left( \mathrm{d}z + P \right) \right] \wedge \Omega_6 \,. \tag{2.36}$$

However, this form does not extend to a global form unless the duality bundle $\mathcal{L}$ is trivial, i.e. the axio-dilaton is constant. To see this note that (2.7) implies that $\mathrm{e}^{\mathrm{i}z}$ transforms as a local section[4] of $K_{\mathcal{M}_6}^{-1} \otimes \mathcal{L}^{-1}$, whereas $\Omega_6$ is a local section of $K_{\mathcal{M}_6}$. The object $\Omega_{(4,0)}$ therefore transforms as a local section of $\mathcal{L}^{-1}$. Since $\mathcal{L}$ admits a global holomorphic section its dual does not, unless $\mathcal{L}$ is trivial. In particular, $\Omega_{(4,0)}$ is not globally defined as a form, when the axio-dilaton varies.

---

[4]We are suppressing the pullbacks in the notation for various bundles.

To circumvent this issue we use the auxiliary elliptic fibration introduced in (2.11), where the complex structure of the elliptic fiber encodes the axio-dilaton. Moreover, we assume that this fibration has a holomorphic section $\sigma : \mathcal{M}_6 \to \mathcal{M}_8^\tau$. Since $\tau$ is preserved by the Killing vector we can construct an elliptic fibration[5] over $Y_7$ by letting the elliptic fiber be constant along the orbits of $\xi$. This gives a 9d space, which we denote by $Y_9^\tau$, endowed with the metric

$$\mathrm{d}s^2(Y_9^\tau) = \mathrm{d}s^2(Y_7) + \mathrm{e}^{B_{10}}\mathrm{d}s^2(\mathbb{E}_\tau) = \eta^2 + \mathrm{e}^{B_{10}}\mathrm{d}s^2\left(\mathcal{M}_8^\tau\right). \tag{2.37}$$

One can think of $Y_9^\tau$ as an elliptic fibration over $Y_7$, with the elliptic fibers being invariant along the Killing vector direction $\xi = 2\partial_z$. The differential forms pull back from $Y_7$ to $Y_9^\tau$, and as usual we conflate the forms with their lifts to avoid notational clutter. We can now define the cone over $Y_9^\tau$ as

$$\mathrm{d}s^2(C(Y_9^\tau)) = \mathrm{d}r^2 + r^2\mathrm{d}s^2(Y_9^\tau). \tag{2.38}$$

This cone admits a natural $SU(5)$ structure, with the $(5,0)$-form locally given by

$$\Omega_{(5,0)} = \mathrm{e}^{\mathrm{i}z}\mathrm{e}^{2B_{10}}r^4\left[\mathrm{d}r - \frac{\mathrm{i}r}{2}\left(\mathrm{d}z + P\right)\right] \wedge \Omega_8. \tag{2.39}$$

The fundamental two-form is exactly the same as in [32] and is not relevant for our purposes. The local holomorphic volume form on $\mathcal{M}_8^\tau$ is

$$\Omega_8 = \mathcal{P} \wedge \Omega_6, \tag{2.40}$$

which satisfies

$$\mathrm{d}\Omega_8 = \mathrm{i}P \wedge \Omega_8. \tag{2.41}$$

The local holomorphic volume form $\Omega_{(5,0)}$ now does extend to a global form, as $\Omega_8$ is a section of $K_{\mathcal{M}_6} \otimes \mathcal{L}$ and the extra $\mathcal{L}$ now cancels with the $\mathcal{L}^{-1}$. In addition, by using (2.41) we can show that the holomorphic volume form is conformally closed

$$\mathrm{d}\Psi = 0, \qquad \Psi \equiv \mathrm{e}^{-2B_{10}}r^{-7}\Omega_{(5,0)}, \tag{2.42}$$

i.e. $C(Y_9^\tau)$ has vanishing first Chern class. We find that $\Psi$ is charged under the R-symmetry vector field

$$\mathcal{L}_\xi\Psi = 2\mathrm{i}\Psi. \tag{2.43}$$

This implies that $\xi$ is a holomorphic vector field, which is paired with the radial vector field under the complex structure $\mathcal{I}(\xi) = -r\partial_r$.

---

[5]This is a fibration in the sense of algebraic geometry, i.e. with a generic fiber being a smooth elliptic curve.

Suppose now that $C(Y_9^\tau)$ admits a holomorphic $U(1)^s$ action, generated by a set of holomorphic vector fields $\partial_{\varphi_i}$, $i = 1, 2, ..., s$. We parametrize the general R-symmetry vector in terms of these holomorphic vector fields

$$\xi = \sum_{i=1}^{s} b_i \partial_{\varphi_i} \,, \tag{2.44}$$

and choose a basis where $\Psi$ has charge 1 under $\partial_{\varphi_1}$ and charge 0 under the remaining generators. This fixes $b_1 = 2$, and leaves the remaining $b_i$, $i = 2, 3, ..., s$ as free variables to be extremized over in $S_F$.

We can now summarize the extremization principle in F-theory: The supersymmetic action $S_F$ in (2.21) is a function of the R-symmetry vector $\xi$ defined in (2.44) and the basic Kähler class $[J_6]$ of $\mathcal{M}_6$. Imposing the flux quantization conditions (2.29) and the associated topological constraints (2.30) and (2.32) relate the R-symmetry parameters ($b_i$ above) and the transverse Kähler class parameters. A putative solution extremizes the supersymmetric action over the remaining free variables. The central charge of the dual SCFT is then computed using (2.26). We shall exemplify this procedure in section 5.

## 3 M/F-Duality and Holographic $\mathcal{I}$-Extremization

The axio-dilaton in F-theory can at times be somewhat obscure, as it is not part of the geometry of the spacetime. To clarify the role of the elliptic fibration, it is often useful to consider a dual M-theory background. For AdS$_3$ solutions, this could either be a dual AdS$_3$ or AdS$_2$ solution of M-theory. In the current framework we will dualize to the latter, which in the field theory corresponds to the circle-reduction to a 1d SCQM. The associated geometric extremization principle, holographic $\mathcal{I}$-extremization, was studied in [32]. In this section we will apply this formalism to the class of geometries that are dual to the F-theory backgrounds and study the extremization principle.

### 3.1 M/F-Duality for AdS-Geometries

To begin with, we will briefly summarize M/F-duality, applied to the F-theory AdS$_3$ geometries discussed in section 2, which are mapped to AdS$_2$ geometries in M-theory.

Any M-theory geometry with an elliptic fibration can be dualized to obtain a corresponding F-theory geometry with varying axio-dilaton, by first reducing to Type IIA along one cycle of the elliptic fibration and subsequently T-dualizing along the second cycle. This approach is valid away from singular fibers, where locally the geometry of the elliptic fiber is

$$\mathrm{d}s^2\left(\mathbb{E}_\tau\right) = \frac{L_{11}^2}{\tau_2}\left((\mathrm{d}x + \tau_1 \mathrm{d}y)^2 + \tau_2^2 \mathrm{d}y^2\right) \,. \tag{3.1}$$

We have introduced an overall M-theory length scale $L_{11}$ and the periodic coordinates $x \sim x + 2\pi\Delta x$ and $y \sim y + 2\pi\Delta y$, where we set $\Delta x = \Delta y$. The M-theory background can be dimensionally reduced on a circle to yield a Type IIA background. Specifically, the two metrics are related as [43]

$$\mathrm{d}s_{11}^2 = L_{11}^2 \left(\frac{l_s}{l_p}\right)^4 \mathrm{e}^{4\phi_{\mathrm{IIA}}/3}(\mathrm{d}x + C_1)^2 + \left(\frac{l_p}{l_s}\right)^2 \mathrm{e}^{-2\phi_{\mathrm{IIA}}/3}\mathrm{d}s_{\mathrm{IIA}}^2 \,, \tag{3.2}$$

where $l_s$ and $l_p$ are the string and 11d Planck lengths, and $\mathrm{d}s_{\mathrm{IIA}}^2$, $\mathrm{e}^{\phi_{\mathrm{IIA}}}$ and $C_1$ are respectively the metric, the fluctuating dilaton and the RR one-form potential of Type IIA. Comparison with (3.1) allows us to immediately identify

$$C_1 = \tau_1 \mathrm{d}y \,, \qquad \mathrm{e}^{4\phi_{\mathrm{IIA}}/3} = \left(\frac{l_p}{l_s}\right)^4 \frac{1}{\tau_2} \,, \qquad \mathrm{d}s_{\mathrm{IIA}}^2 = L_{11}^2 \left(\frac{l_s}{l_p}\right)^2 \mathrm{e}^{2\phi_{\mathrm{IIA}}/3}\tau_2 \mathrm{d}y^2 + \mathrm{d}s_9^2 \,, \tag{3.3}$$

where $\mathrm{d}s_9^2$ is the metric on the 9d space of the Type IIA geometry orthogonal to the $y$ circle. Dimensionally reducing the M-theory action to that of Type IIA (here it is sufficient to consider the Ricci scalar term) fixes the period of the circle to be

$$L_{11}\Delta x = \frac{l_p^3}{l_s^2} \,, \tag{3.4}$$

where we have used $16\pi G_{11} = (2\pi)^8 l_p^9$ and $16\pi G_{10} = (2\pi)^7 l_s^8$ for the 11d and 10d Newton constants, respectively. Hence, we can express the volume of the elliptic fiber in terms of fundamental length scales as

$$\mathrm{vol}\left(\mathbb{E}_\tau\right) = (2\pi\Delta x)^2 = \frac{(2\pi)^2 l_p^6}{L_{11}^2 l_s^4} \,. \tag{3.5}$$

Carrying out T-duality along the $y$ circle results in

$$R_{\mathrm{IIB}} = \frac{l_s^2}{L_{11}\Delta y} = \frac{l_s^4}{l_p^3} \,, \qquad C_0 = (C_1)_y = \tau_1 \,, \qquad \mathrm{e}^{\phi_{\mathrm{IIB}}} = \frac{l_s}{\frac{l_s}{l_p}L_{11}\Delta y\,\mathrm{e}^{\phi_{\mathrm{IIA}}/3}\sqrt{\tau_2}}\mathrm{e}^{\phi_{\mathrm{IIA}}} = \frac{1}{\tau_2} \,. \tag{3.6}$$

This then identifies $\tau = \tau_1 + \mathrm{i}\tau_2 = C_0 + \mathrm{i}\,\mathrm{e}^{-\phi_{\mathrm{IIB}}}$.

Applied to the AdS$_3$ F-theory geometries of section 2, the key observation is that we dualize along the AdS direction by first writing AdS$_3$ as a circle fibration over AdS$_2$ [44]

$$\mathrm{d}s^2\left(\mathrm{AdS}_3\right) = \frac{1}{4}\left(-r^2\mathrm{d}t^2 + \frac{\mathrm{d}r^2}{r^2} + (\mathrm{d}\phi + a_1)^2\right) = \frac{1}{4}\mathrm{d}s^2\left(\mathrm{AdS}_2\right) + \frac{1}{4}(\mathrm{d}\phi + a_1)^2 \,, \tag{3.7}$$

where $\phi \sim \phi + \Delta\phi$ is the circle coordinate and $a_1 = r\mathrm{d}t$ so that $\mathrm{d}a_1 = \mathrm{vol}_{\mathrm{AdS}_2}$. The F-theory metric can then be written as

$$\mathrm{d}s_{10}^2 = L_{11}^2 \mathrm{e}^{-B_{11}/2}\left[\mathrm{d}s^2\left(\mathrm{AdS}_2\right) + (\mathrm{d}\phi + a_1)^2 + (\mathrm{d}z + P)^2 + \mathrm{e}^{B_{11}}\mathrm{d}s^2\left(\mathcal{M}_6\right)\right] \,, \tag{3.8}$$

where we have taken the M/F-theory length scales and warp factors to be related by

$$L_{10} = \sqrt{2}L_{11}, \qquad e^{B_{10}} = \frac{1}{4}e^{B_{11}}.$$ (3.9)

T-duality along the $\phi$ direction results in

$$
\begin{aligned}
ds^2_{\text{IIA}} &= L^2_{11}\sqrt{\tau_2}\, e^{B_{11}/2}d\phi^2 + L^2_{11}\frac{e^{-B_{11}/2}}{\sqrt{\tau_2}}\left[ds^2\left(\text{AdS}_2\right) + (dz + P)^2 + e^{B_{11}}ds^2\left(\mathcal{M}_6\right)\right], \\
e^{-2\phi_{\text{IIA}}} &= \frac{l^6_p}{l^6_s}\tau_2^{3/2}e^{-B_{11}/2}, \\
H &= L^2_{11}d\phi \wedge \text{vol}_{\text{AdS}_2}, \\
F_2 &= L_{11}d\tau_1 \wedge d\phi, \\
F_4 &= \frac{1}{2}L^3_{11}\text{vol}_{\text{AdS}_2} \wedge F.
\end{aligned}
$$ (3.10)

Finally, we uplift to M-theory using the metric in (3.2). We find that the M-theory geometries dual to the AdS$_3$ F-theory geometries in (2.9) are

$$
\begin{aligned}
ds^2_{11} &= L^2_{11}e^{-2B_{11}/3}\left[ds^2\left(\text{AdS}_2\right) + (dz + P)^2 + e^{B_{11}}ds^2\left(\mathcal{M}^\tau_8\right)\right], \\
G_4 &= L^3_{11}\text{vol}_{\text{AdS}_2} \wedge \left[-J_8 + d\left(e^{-B_{11}}\left(dz + P\right)\right)\right], \\
dP &= \rho_8, \\
e^{B_{11}} &= \frac{1}{2}R_8,
\end{aligned}
$$ (3.11)

where $J_8$, $\rho_8$ and $R_8$ denote the Kähler form, Ricci form, and Ricci scalar of the Kähler four-fold $\mathcal{M}^\tau_8$. This is exactly the space introduced in (2.11) with metric (2.37), which in M-theory forms part of the physical spacetime. Its Ricci form and scalar are related to the corresponding $\mathcal{M}_6$ quantities as

$$\rho_8 = \rho_6 - i\mathcal{P} \wedge \bar{\mathcal{P}}, \qquad R_8 = R_6 - 2|\mathcal{P}|^2.$$ (3.12)

Notice that the duality determines the period of the $\phi$ circle in terms of fundamental length scales to be

$$\frac{L_{11}\Delta\phi}{2\pi} = \frac{l^4_s}{l^3_p}.$$ (3.13)

## 3.2 Holographic $\mathcal{I}$-Extremization

In this section we will briefly summarize the central aspects of the general version of holographic $\mathcal{I}$-extremization, before specializing to elliptically fibered M-theory geometries. We refer the reader to [32] for a complete account.

The extremization principle applies to any supersymmetric M-theory AdS$_2$ geometry with electric four-form flux

$$ds_{11}^2 = L_{11}^2 e^{-2B_{11}/3} \left[ ds^2 \left( \text{AdS}_2 \right) + ds^2 \left( Y_9^\tau \right) \right] ,$$
$$G_4 = L_{11}^3 \text{vol}_{\text{AdS}_2} \wedge F ,$$

(3.14)

where the compact internal space $Y_9^\tau$ admits a natural unit length Killing vector $\xi$. The leaf spaces of the transverse foliation $\mathcal{F}_\xi$ admit a Kähler structure. Following the notation in section 3.1 we denote that transverse space by $\mathcal{M}_8^\tau$. Note that this internal space differs from the F-theory compact space with the auxiliary elliptic fibration by the normalization of the Killing vector. A detailed comparison of the metrics and normalizations in M- versus F-theory is included in appendix A. The warp factor $B_{11}$ and closed two-form $F$ are defined on the internal space $Y_9^\tau$. These supersymmetric geometries can be put on-shell by imposing the condition

$$\Box_8 R_8 = \frac{1}{2} R_8^2 - (R_8)_{\mu\nu} (R_8)^{\mu\nu}$$

(3.15)

on the transverse space. Clearly, the M-theory duals derived in the previous section belong to this class of theories. The geometric dual of $\mathcal{I}$-extremization is formulated as follows. For an M-theory supergravity background to be consistent, the $G_4$-flux has to be quantized before extremizing, which is well-defined if we impose the topological restriction

$$H^2(Y_9^\tau, \mathbb{R}) = H_B^2(\mathcal{F}_\xi)/[\rho_8]$$

(3.16)

and constraint equation

$$\int_{Y_9^\tau} \eta \wedge \rho_8^2 \wedge \frac{J_8^2}{2} = 0 .$$

(3.17)

Then flux quantization over all seven-cycles $\widetilde{S}_I \in H_7(Y_9^\tau, \mathbb{Z})$ is given by

$$\nu_4 N_I^M = \int_{\widetilde{S}_I} \eta \wedge \rho_8 \wedge \frac{J_8^2}{2} ,$$

(3.18)

where we have introduced the positive constant

$$\nu_4 = \frac{(2\pi l_p)^6}{L_{11}^6} .$$

(3.19)

An analogous cone $C(Y_9^\tau)$ to the one presented in section 2.4 can be constructed for the present M-theory geometries. This cone likewise has a globally defined holomorphic $(5,0)$-form, which allows for a parametrization of the M-theory R-symmetry vector in terms of a $U(1)^s$ action on the cone, as in (2.44). Note that the holomorphic $(5,0)$-form on $C(Y_9^\tau)$ has charge 1 under

the R-symmetry vector field, so that the first coefficient in the parametrization is $b_1 = 1$ (again, see appendix A for a discussion of the difference in normalization of the R-symmetry vector in M- versus F-theory).

Having fixed a complex cone $C(Y_9^\tau)$ and imposed the constraint equation (3.17) and flux quantization (3.18), the supersymmetric action

$$S_M = \int_{Y_9^\tau} \eta \wedge \rho_8 \wedge \frac{J_8^3}{3!} , \tag{3.20}$$

can be extremized with respect to the remaining variables in $\xi$ and $[J_8]$. This is a necessary condition for the geometry $Y_9^\tau$ to be on-shell, i.e. to satisfy the master equation (3.15). The effective $\mathrm{AdS}_2$ Newton constant of such an on-shell solution is then

$$\frac{1}{G_2} = \frac{8(2\pi)^2}{\nu_4^{3/2}} S_M \bigg|_{\text{on-shell}} . \tag{3.21}$$

### 3.3 M-Theory Supersymmetric Action for Elliptic Fibrations

Finally, we specialize the M-theory geometries to those with F-theory duals, i.e. $\mathcal{M}_8^\tau$ is an elliptic fibration over a base $\mathcal{M}_6$ with a section $\sigma : \mathcal{M}_6 \to \mathcal{M}_8^\tau$. We are interested in determining how the flux quantization conditions (3.18) and supersymmetric action (3.20) depend on data of the base $\mathcal{M}_6$. For this purpose we will here focus on on-shell solutions, which allows us to assume a choice of a regular Killing vector. This in turn ensures that the transverse Kähler space $\mathcal{M}_8^\tau$ is a smooth manifold. We will return to the extremization problem in the subsequent sections.

The Shioda-Tate-Wazir theorem for elliptically fibered Kähler manifolds [45] asserts[6] that we can decompose the (cohomology class of the) Kähler form on $\mathcal{M}_8^\tau$ as

$$J_8 = k_0 \omega_0 + \sum_\alpha k_\alpha \omega_\alpha + \sum_i k_i \omega_i \equiv \sum_I k_I \omega_I . \tag{3.22}$$

This decomposition corresponds to three divisor classes, which generate the Picard group of $\mathcal{M}_8^\tau$. These are: the divisor corresponding to the section $\sigma$ with its dual (1,1)-form $\omega_0$, the pullback divisors $C_\alpha$ with dual forms denoted by $\omega_\alpha$, and finally the resolution divisors (also referred to as Cartan divisors) $D_i$ with dual forms $\omega_i$. For a more thorough discussion see [1]. Note that we do not require the Kähler parameters $k_I$ to be integers; rather they are real numbers, which will ultimately be determined by the flux integers. Moreover, the Killing

---

[6]For this to be true we need to impose some topological restrictions, namely $h^{1,0}(\mathcal{M}_8^\tau) = h^{2,0}(\mathcal{M}_8^\tau) = 0$. Also we assume for simplicity that there are no extra sections, i.e. the Mordell Weil group is trivial. From now on we assume this to hold.

vector is assumed to be regular, implying a smooth $\mathcal{M}_8^\tau$. We assume for simplicity that the elliptic fibration is a smooth Weierstrass model and thus only has Kodaira type $I_1$ fibers and no resolution divisors.

With the expansion (3.22) the supersymmetric action (3.20) becomes

$$S_M = \sum_{IJK} \frac{k_I k_J k_K}{3!} \int_{Y_9^\tau} \eta \wedge \rho_8 \wedge \omega_I \wedge \omega_J \wedge \omega_K \,. \tag{3.23}$$

The integral in $S_M$ can be pushed down to an intersection on the base using adjunction

$$c_1(\mathcal{M}_8^\tau) = c_1(\mathcal{M}_6) - c_1(\mathcal{L}) \,. \tag{3.24}$$

Furthermore, since the Killing vector is regular, we can integrate out the circle direction, which we take to have period $2\pi\ell$, and write the supersymmetric action as

$$S_M = (2\pi)^2 \ell \sum_{IJK} \frac{k_I k_J k_K}{3!} \int_{\mathcal{M}_8^\tau} (c_1(\mathcal{M}_6) - c_1(\mathcal{L})) \wedge \omega_I \wedge \omega_J \wedge \omega_K \,. \tag{3.25}$$

We define the intersection numbers

$$C_{IJK} \equiv (c_1(\mathcal{M}_6) - c_1(\mathcal{L})) \cdot C_I \cdot C_J \cdot C_K = \int_{\mathcal{M}_8^\tau} (c_1(\mathcal{M}_6) - c_1(\mathcal{L})) \wedge \omega_I \wedge \omega_J \wedge \omega_K \,. \tag{3.26}$$

Using the intersection identity

$$\sigma \cdot_{\mathcal{M}_8^\tau} (\sigma + c_1(\mathcal{L})) = 0 \,, \tag{3.27}$$

a short computation shows that

$$\begin{aligned}
C_{000} &= (c_1(\mathcal{M}_6) - c_1(\mathcal{L})) \cdot c_1(\mathcal{L}) \cdot c_1(\mathcal{L}) \,, \\
C_{00\alpha} &= -(c_1(\mathcal{M}_6) - c_1(\mathcal{L})) \cdot c_1(\mathcal{L}) \cdot C_\alpha \,, \\
C_{0\alpha\beta} &= (c_1(\mathcal{M}_6) - c_1(\mathcal{L})) \cdot C_\alpha \cdot C_\beta \,, \\
C_{\alpha\beta\gamma} &= 0 \,,
\end{aligned} \tag{3.28}$$

which are manifestly intersection numbers on the base $\mathcal{M}_6$. Then the supersymmetric action specialized to elliptic fibrations is given in terms of intersection numbers on the base as

$$S_M = (2\pi)^2 \ell \sum_{IJK} \frac{k_I k_J k_K}{3!} C_{IJK} \,. \tag{3.29}$$

The flux quantization conditions (3.18) specialized to elliptic fibrations become

$$\nu_4 N_I^M = (2\pi)^2 \ell \sum_{JK} \frac{k_J k_K}{2} C_{IJK} \,. \tag{3.30}$$

Finally, observe that the Kähler parameter $k_0$ of the elliptic fiber is exactly the volume of a (non-singular) fiber

$$\text{vol}(\mathbb{E}_\tau) = \int_{\mathbb{E}_\tau} J_8 = k_0 \int_{\mathbb{E}_\tau} \omega_0 = k_0 \,. \tag{3.31}$$

From the discussion of the M/F-duality, specifically using (3.5), we find that $k_0$ is expressed in terms of fundamental lengths as

$$k_0 = \frac{(2\pi)^2 l_p^6}{L_{11}^2 l_s^4} \,. \tag{3.32}$$

## 4  $\mathcal{I}/c$-Extremization

We will now compare the extremization procedures in M/F-theory. We will first provide the map between the two geometric extremization procedures, and then discuss the dual field theory.

### 4.1  Geometry

What we have argued so far is that an F-theory AdS$_3$ geometry is characterized by the complex geometry of the internal space $\mathcal{M}_6$ and the axio-dilaton profile. They are conveniently thought of here in terms of the complex cone $C(Y_9^\tau)$, which is a $\mathbb{C}^*$ fibration over an elliptically fibered base $\mathcal{M}_8^\tau$. An on-shell solution is ultimately determined by imposing a topological constraint, as well as a choice of quantized flux numbers $N_\alpha^F \in \mathbb{Z}$, where $\alpha = 1, \ldots, \dim H_5(Y_7, \mathbb{R})$, as these fix the Kähler class parameters of the internal space geometry. Such a solution is then dual to a 2d $(0,2)$ SCFT living on the conformal boundary of AdS$_3$, for example as written in the usual Poincaré slicing. The holographic central charge $c_{\text{sugra}}$ of this theory is computed using equation (2.26).

Associated to any such F-theory solution is a different global form of AdS$_3$, which is a circle bundle over AdS$_2$, as in (3.7). Topologically the circle fibration is trivial, with the fiber coordinate $\phi$ having period $\Delta\phi$, which *a priori* is arbitrary. Since the size of the $\phi$ circle in the AdS$_3$ is bounded it becomes part of the internal space, and the remaining conformal boundary is 1-dimensional. This implies that the associated solutions have an interpretation as holographic duals to 1d SCQM.

T-dualizing along this circle and uplifting to M-theory, the circle becomes part of the internal space of the M-theory geometry and, together with the circle introduced in the uplift from Type IIA to M-theory, it makes up the elliptic fiber $\mathbb{E}_\tau$ with volume $k_0$. The M-theory AdS$_2$ geometries obtained in this way are determined by the complex geometry

of the internal space $\mathcal{M}_8^\tau$. An analogous cone construction $C(Y_9^\tau)$ exists for the M-theory geometries [32], which provides a parametrization of the R-symmetry vector. Finding on-shell M-theory solutions amounts to imposing a topological constraint and a choice of flux numbers $N_I^M \in \mathbb{Z}$, where $I = 1, \ldots, \dim H_7(Y_9^\tau, \mathbb{R})$, which fix the Kähler class parameters of the internal complex geometry. The effective AdS$_2$ Newton constant is then computed as in (3.21).

The two supergravity duals each contain a set of parameters that are mapped to each other through the duality. On either side, the flux quantization conditions come with a dimensionless combination of length scales characteristic of each theory, namely $\nu_3$ in F-theory and $\nu_4$ in M-theory. Furthermore, on the F-theory side we have the circle length $\Delta\phi$ as an *a priori* free parameter, and on the M-theory side we have the fiber volume $k_0$. These parameters are given in terms of fundamental length scales as

$$
\text{F-theory/IIB}: \quad
\begin{cases}
\nu_3 = \dfrac{2(2\pi l_s)^4}{L_{10}^4} \\[2mm]
\dfrac{\Delta\phi}{2\pi} = \dfrac{\sqrt{2}l_s^4}{L_{10}l_p^3}
\end{cases}
\qquad
\text{M-theory}: \quad
\begin{cases}
\nu_4 = \dfrac{(2\pi l_p)^6}{L_{11}^6} \\[2mm]
k_0 = \dfrac{(2\pi)^2 l_p^6}{L_{11}^2 l_s^4}
\end{cases}
\tag{4.1}
$$

With $L_{11} = L_{10}/\sqrt{2}$, we find the following relation

$$
\Delta\phi = \frac{\sqrt{\nu_4}}{k_0} \, .
\tag{4.2}
$$

As T-duality inverts the radius of the circle, $\Delta\phi$ is indeed expected to be inversely related to the volume of the elliptic fiber. Given such an M-theory geometry, we can trace through the duality in the other direction by taking the F-theory limit, corresponding to shrinking the elliptic fiber to zero size, $k_0 \to 0$. This in turn takes $\Delta\phi \to \infty$, decompactifying the $\phi$ circle.

Any solution to the topological constraint together with some configuration of flux numbers makes for a perfectly consistent and physical M-theory solution. However, in this paper we are not interested in a generic M-theory solution; rather, we wish to find the ones *with F-theory duals*, and the map that takes us from one to the other.

For this purpose, it is instructive to compare Kähler classes on the two sides of the duality, focusing on the $k_0$ dependence on the M-theory side, since the F-theory limit takes $k_0$ to zero. In other words, we concentrate on the contributions coming from the volume of the elliptic fibration, which forms part of the physical data in M-theory, and ceases to have a physical interpretation in F-theory. Consider again the decomposition of the 8d Kähler form $J_8$ in (3.22)

$$
J_8 = k_0 \omega_0 + \sum_\alpha k_\alpha \omega_\alpha \, .
\tag{4.3}
$$

Recall that $\omega_\alpha$ are pullbacks from the base $\mathcal{M}_6$ and together with $\omega_0$ generate the second integral cohomology of $\mathcal{M}_8^\tau$. Once the M-theory topological constraint is imposed and the fluxes are properly quantized, the parameters $k_\alpha$ depend implicitly on the size of the elliptic fiber $k_0$. We will see this in examples in later sections. We denote the Kähler parameters of $J_6$ by $\mathsf{k}_\alpha$ such that

$$J_6 = \sum_\alpha \mathsf{k}_\alpha \omega_\alpha \,. \tag{4.4}$$

The requirement for mapping a specific M-theory solution to its F-theory dual is that the Kähler class on $\mathcal{M}_8^\tau$ should match that of $\mathcal{M}_6$ in the F-theory limit, i.e.

$$J_6 = \lim_{k_0 \to 0} J_8 = \lim_{k_0 \to 0} \sum_\alpha k_\alpha \omega_\alpha \,. \tag{4.5}$$

In geometric terms we are collapsing the elliptic fiber, while keeping the volume of the total space bounded. The metric on $\mathcal{M}_8^\tau$, with Kähler form $J_8$, then under appropriate convergence conditions tends to a (singular) metric on $\mathcal{M}_6$, with Kähler form $J_6$. This implies that the 8d and 6d Kähler parameters are related by

$$k_\alpha = \mathsf{k}_\alpha + \mathcal{O}(k_0) \,. \tag{4.6}$$

This ansatz for the decomposition of the 8d Kähler form results in an M-theory topological constraint equation, which can be expanded order by order in $k_0$ to give

$$k_0 \int_{Y_7} \eta \wedge (c_1(\mathcal{M}_6) - c_1(\mathcal{L}))^2 \wedge J_6 + \mathcal{O}(k_0^2) = 0 \,. \tag{4.7}$$

The lowest order term is exactly the constraint equation for the F-theory geometries that was independently derived in section 2.3. Since this equation must be satisfied order by order in $k_0$, the F-theory constraint equation is thus built into its dual M-theory solution by imposing that $J_8$ satisfy (4.5). Requiring the higher order terms to vanish constrains the form of (4.6).

For every M-theory flux integer $N_\alpha^M \in \mathbb{Z}$ there exists an F-theory flux integer $N_\alpha^F \in \mathbb{Z}$, where the M-theory seven-cycle is exactly the corresponding F-theory five-cycle with the elliptic fibration. The requirement (4.5) ensures that in dual solutions these flux configurations match on the nose, i.e. we have $N_\alpha^M = N_\alpha^F \equiv N_\alpha \in \mathbb{Z}$. In a sense, this condition expresses the fact that every D3-brane is simply converted to an M2-brane.

When determining an on-shell F-theory solution, imposing the $N_\alpha$ flux quantization conditions and the topological constraint determine the complex geometry of $Y_7$ by fixing the R-symmetry vector and the Kähler parameters of $\mathcal{M}_6$. In M-theory there is an additional

distinguished flux integer $N_0^M \equiv N_0$, which has no F-theory analog, as it arises from the section $\sigma$ of the elliptic fibration, i.e.

$$\nu_4 N_0 = \int_{Y_7} \eta \wedge \rho_8 \wedge \frac{J_8^2}{2} \,. \tag{4.8}$$

The distinguished flux integer does not map to any flux integer present in F-theory; rather, expanding in orders of $k_0$, we find that its leading contribution is determined by the Type IIB $\phi$ circle length and the central charge as

$$N_0 = \left(\frac{\Delta\phi}{2\pi}\right)^2 \frac{c_{\text{sugra}}}{24} + \mathcal{O}(k_0) \,. \tag{4.9}$$

This additional flux quantization condition is matched by the extra Kähler parameter $k_0$ in the compact space of the M-theory geometry. In practice, as we shall see in examples, fixing this additional flux number then fixes the period $\Delta\phi$. Hence, imposing the M-theory topological constraint and flux quantization conditions determines the decomposition of $J_8$ and the internal space geometry $Y_9^\tau$.

The map from holographic $\mathcal{I}$-extremization in M-theory to $c$-extremization in F-theory is completed by considering the relation between the two actions. *Before* imposing the topological constraint or flux quantization, the dual supersymmetric actions are related as

$$S_M = 2k_0 S_F + \mathcal{O}(k_0^2) \,. \tag{4.10}$$

The factor of 2 comes from the relative rescaling of the Killing one-form $\eta$ (see appendix A for a discussion of the relative normalizations in M- versus F-theory). The on-shell central charge of the 2d SCFT is then formally related to the AdS$_2$ Newton constant by

$$\frac{1}{G_2} = \frac{\Delta\phi}{3} c_{\text{sugra}} + \mathcal{O}(k_0) \,. \tag{4.11}$$

The reason this should only be read as a formal expression for the AdS$_2$ Newton constant is that the $N_0$ flux quantization condition has not been imposed, thus leaving in factors of $k_0$. Since a Kähler parameter of the internal space still appears explicitly in the equations, it cannot be understood as a physical quantity. From the above, we can thus conclude that *holographic $\mathcal{I}$-extremization in M-theory does not in general equal holographic c-extremization in F-theory.* In other words, extremizing $1/G_2$ does not necessarily correspond to finding an extremum for $c_{\text{sugra}}$.

This result generalizes the relation derived in [46], where the AdS$_2$ is considered as arising directly in Type IIB by writing AdS$_3$ as the total space of a circle fibration. The effective

Newton constants are then related by dimensional reduction on this circle

$$G_3 = \frac{\Delta\phi}{2}G_2 \,, \tag{4.12}$$

where the factor of $1/2$ here arises as the length of the $\phi$ circle in the AdS$_3$ metric (3.7). Equation (4.11) takes into account corrections from the 7-branes and exactly reduces to the supergravity result in (4.12) when the elliptic fibration is trivial.

Interestingly, for many cases that we study later in this paper, the $\mathcal{O}(k_0)$ terms are in fact absent in (4.11), even for a non-trivial elliptic fibration, so that (4.12) holds exactly. This is true for all the toric examples in section 5. As a proof of concept, we therefore also consider a known set of solutions, the universal twist solutions with elliptic three-fold factor, in section 6, which do have non-zero subleading terms.

### 4.2 Field Theory

Finally, we comment on the physical interpretation of (4.11) in terms of the holographically dual field theories. First recall how the two field theory duals are constructed. On the F-theory side, the dual field theories are realized on D3-branes along $\mathbb{R}^{1,1} \times C$, where $C$ are curves in F-theory compactifications, above which the axio-dilaton profile is non-trivial. This induces a varying coupling $\tau$ of the 4d gauge theory on the D3-branes, and the 2d $(0,2)$ field theory along $\mathbb{R}^{1,1}$ acquires a dependence on the $U(1)_D$ duality line bundle $\mathcal{L}$ [13–16, 21]. T-duality along a circle in the D3-brane world-volume gives rise to a configuration of D2-branes, which uplift in M-theory to M2-branes wrapped on the curves $C$, i.e. the M2-branes realize a 1d SCQM.

While AdS$_2$ holography is still very much under development, it is natural to identify minus the logarithm of the partition function of the 1d theory with the renormalized supergravity action. As shown in [32] we may thus identify

$$\log Z_{1d} = \frac{1}{4G_2} \,. \tag{4.13}$$

If we consider the 1d SCQM as arising directly from a circle reduction of the 2d (0,2) SCFT, or, equivalently, from duality with M-theory on a trivially fibered torus, then we can use (4.12) and the standard Brown-Henneaux relation [42] to deduce that

$$\log Z_{1d} = \frac{1}{4G_2} = \frac{\Delta\phi}{8G_3} = \frac{\Delta\phi}{12}c_{\text{sugra}} \,. \tag{4.14}$$

Of course this is precisely equation (4.11), without the $\mathcal{O}(k_0)$ correction terms. The partition function on the left hand side of (4.13) is defined by putting the 1d SCQM on a circle. On

the other hand, we have also effectively reduced from 2d to 1d on the $\phi$ circle. Physically one might then anticipate some relation between the 1d partition function and the 2d partition function, where the 2d $(0,2)$ theory is put on a torus $T^2$. We note that this is indeed precisely the case: putting a 2d CFT on a torus leads to a Casimir energy contribution to the partition function

$$Z_{T^2}(\text{Casimir}) = \exp\left(\frac{r_1}{r_2}\frac{c}{12}\right), \qquad (4.15)$$

where $r_1, r_2$ are the lengths of the circles in the $T^2$, and $c$ is the central charge. We should then also recall that $\Delta\phi$ is dimensionless, but may be written as

$$\Delta\phi = \frac{2\pi R_{\text{IIB}}}{L_{11}}, \qquad (4.16)$$

where $R_{\text{IIB}}$ is the dimensionful Type IIB $\phi$ circle length and $L_{11}$ is the overall dimensionful length scale in M-theory. The right hand side of (4.14) may then be identified with (the logarithm of) this Casimir contribution to the $T^2$ partition function. Recall here that in the M-theory solution $\Delta\phi$ depends on the additional M-theory flux number $N_0$, while the central charge $c_{\text{sugra}}$ depends only on the F-theory data, which does not include $N_0$. In the above identification, the extra parameter $N_0$ determines, via $\Delta\phi$, the geometry of the $T^2$ on which the 2d $(0,2)$ SCFT is placed. Notice then that the 2d $(0,2)$ theory itself does not depend on the integer $N_0$, while the 1d SCQM that it reduces to does depend on $N_0$. It would be interesting to understand this in more detail, and in particular whether the integer $N_0$ has a simple 1d interpretation. In addition, a study of the supersymmetric Casimir energy [47], its $S^1$-reduction, and the holographic duals, would be of interst and may shed light on subleading corrections.

We will exemplify these general insights by considering several classes of solutions in the next two sections. In section 5 we study M/F-theory dual holographic setups, where the relation (4.11) holds precisely, without any $\mathcal{O}(k_0)$ corrections. We contrast this in section 6, where we study solutions where there are non-trivial corrections as predicted by (4.11). The key difference between these two sets of solutions is that in the former, the elliptic fiber is restricted to a complex curve, whereas in the latter the fibration is non-trivial over a complex surface, which results for instance in non-trivial terms of the type $c_1(\mathcal{L})^2$, which contribute the higher order terms in $k_0$.

## 5 Toric Fibrations over a Curve

In this section, we consider a class of toric geometries fibered over a complex curve or an elliptically fibered surface, where we can derive explicit formulas for the off-shell M/F-theory

extremization problem. We show that, for these geometries, $\mathcal{I}$- and $c$-extremization are equivalent without any corrections in $k_0$, the volume of the elliptic fiber. Moreover, we apply the formalism to the cases referred to in the literature as the *universal* and *baryonic twists*.

## 5.1 Toric Fibration

We are interested in geometries where the compact part of the space consists of a toric five-manifold fibered over either, in the case of F-theory, a Riemann surface $\Sigma$ with genus $g$ or, in M-theory, an elliptic surface

$$
\begin{array}{ccc}
\mathbb{E}_\tau & \longhookrightarrow & B_4^\tau \\
& & \downarrow \\
& & \Sigma
\end{array}
\tag{5.1}
$$

We start with a review of the properties of the toric fiber, which we denote by $Y_5$. We require that the cone $C(Y_5)$ is complex and Kähler, i.e. that $Y_5$ is Sasaki, and that $C(Y_5)$ admits a global holomorphic (3,0)-form. Such cones are called Gorenstein and the geometry of such toric Kähler cones has been extensively studied in e.g. [34, 48]. For our purposes, the essential feature of the fibered toric geometries is that all relevant quantities turn out to be expressible in terms of (derivatives of) a *master volume* of the fiber

$$
\mathcal{V} \equiv \int_{Y_5} \eta \wedge \frac{\omega^2}{2} \, ,
\tag{5.2}
$$

where $\omega$ is the Kähler form on the space transverse to $\eta$ in $Y_5$. Moreover, for a fixed Gorenstein toric Kähler cone $C(Y_5)$, there is a simple explicit expression for $\mathcal{V}$ in terms of the toric data. The master volume is a function of the inward pointing primitive normals to the $d \geq 3$ facets of the polyhedral cone $v_a \in \mathbb{Z}^3$, with $a = 1, ..., d$ , as well as the transverse Kähler parameters $\lambda_a$ and trial R-symmetry vector $\vec{b} = (b_1, b_2, b_3)$. The master volume is given by

$$
\mathcal{V}(\vec{b}, \{\lambda_a\}, \{\vec{v}_a\}) = \frac{(2\pi)^3}{2} \sum_{a=1}^d \lambda_a \frac{\lambda_{a-1}[\vec{v}_a, \vec{v}_{a+1}, \vec{b}] - \lambda_a[\vec{v}_{a-1}, \vec{v}_{a+1}, \vec{b}] + \lambda_{a+1}[\vec{v}_{a-1}, \vec{v}_a, \vec{b}]}{[\vec{v}_{a-1}, \vec{v}_a, \vec{b}][\vec{v}_a, \vec{v}_{a+1}, \vec{b}]} \, .
\tag{5.3}
$$

Here $[\cdot, \cdot, \cdot]$ denotes a $3 \times 3$ determinant, and we cyclically order $\vec{v}_0 = \vec{v}_d$, $\vec{v}_{d+1} = \vec{v}_1$, with similar identifications for the $\lambda_a$. Note that two of the Kähler class parameters are redundant, so that $\mathcal{V}$ is effectively only a function of $d - 2$ of the $d$ Kähler class parameters $\lambda_a$.

## 5.2 F-Theory $c$-Extremization for Toric Fibrations

In this section, we consider the fibration of $Y_5$ over a Riemann surface $\Sigma$, and derive the F-theory $c$-extremization equations specialized to this class of geometries. The fibration of

$Y_5$ over $\Sigma$ can be parametrized as follows. The toric manifold is equipped with an isometric $U(1)^3$ action, generated by a set of holomorphic vector fields $\partial_{\varphi_i}, i = 1, 2, 3$. We choose three line bundles $\mathcal{O}(n_i)$ on the Riemann surface so that, topologically, the compactification space is defined to be the total space of the associated bundle

$$Y_7 = \mathcal{O}(\vec{n}) \times_{U(1)^3} Y_5 \,. \tag{5.4}$$

For simplicity we shall assume that the axio-dilaton varies only over the Riemann surface $\Sigma$. That is, taking the F-theory perspective, the variation of the axio-dilaton is captured by an auxiliary elliptic fibration as in (2.11), where the total space that we will consider is

$$\begin{array}{ccc}
 & & Y_9^\tau \\
 & \swarrow & \downarrow{\scriptstyle p^*(\pi)} \\
\mathbb{E}_\tau \longleftrightarrow B_4^\tau & & Y_7 \\
{\scriptstyle \sigma}\big\uparrow\big\downarrow{\scriptstyle \pi} & \swarrow {\scriptstyle p} & \\
\Sigma & &
\end{array} \tag{5.5}$$

Recall that the manifold $Y_9^\tau$ is obtained by pulling back the elliptic fibration $\pi$ to $Y_7$ as in section 2.4. The existence of a global holomorphic $(5,0)$-form on $C(Y_9^\tau)$ places certain restrictions on $\vec{n}$. We may construct such a global $(5,0)$-form by first noting that $C(Y_5)$ admits a global $(3,0)$-form $\Omega_{(3,0)}$. The $(3,0)$-form has an explicit $e^{i\varphi_1}$ dependence, since it has R-charge 2. On $B_4^\tau$ there is a local $(2,0)$-form $\Xi_{(2,0)}$, which is a local section of $K_{B_4^\tau}$. We have

$$K_{B_4^\tau} = K_\Sigma \otimes \mathcal{L} \tag{5.6}$$

where $\mathcal{L}$ is the duality line bundle, whose connection depends on the variation of the axio-dilaton as introduced in section 2 and

$$\deg(K_{B_4^\tau}) = 2g - 2 + \deg \mathcal{L} \,. \tag{5.7}$$

The holomorphic volume form on $C(Y_9^\tau)$ is constructed as

$$\Omega_{(5,0)} = \Omega_{(3,0)} \wedge \Xi_{(2,0)} \,, \tag{5.8}$$

where $\Omega_{(3,0)}$ is twisted over $\Sigma$ as in (5.4). Since $e^{i\varphi_1}$ is a section of $\mathcal{O}(n_1)$, we can ensure that $\Omega_{(5,0)}$ is a global non-vanishing form by taking

$$n_1 = 2 - 2g - \deg \mathcal{L} \,. \tag{5.9}$$

The twist is implemented at the level of the forms by introducing a connection $A_i$ on each $\mathcal{O}(n_i)$ with curvature $F_i = \mathrm{d}A_i$. The curvatures satisfy

$$\int_\Sigma \frac{F_i}{2\pi} = n_i \in \mathbb{Z}\,. \tag{5.10}$$

The fibration in (5.4) amounts to making the replacements

$$
\begin{aligned}
\eta &\to \eta_{\text{twist}} \equiv \eta + 2\sum_{i=1}^3 w_i A_i\,, \\
\omega &\to \omega_{\text{twist}} \equiv \omega + \sum_{i=1}^3 \left(\mathrm{d}x_i \wedge A_i + x_i F_i\right)\,, \\
J_6 &\to J_{6\text{twist}} = \omega_{\text{twist}} + A\mathrm{vol}_\Sigma\,,
\end{aligned}
\tag{5.11}
$$

where $w_i$ are the moment map coordinates restricted to $Y_5$ and $x_i$ are global functions on $Y_5$ invariant under the $U(1)^3$ action (see [48] for further details). Note that the frequently appearing combination

$$[\rho_6 - 2\pi c_1\left(\mathcal{L}\right)] \to [\rho_6 - 2\pi c_1\left(\mathcal{L}\right)]_{\text{twist}} = [b_1 \mathrm{d}\eta_{\text{twist}}] \tag{5.12}$$

also obtains an $A_i$ dependence under the twist. With these replacements the supersymmetric action is

$$
\begin{aligned}
S_F &= \int_{Y_7} \eta_{\text{twist}} \wedge [\rho_6 - 2\pi c_1(\mathcal{L})]_{\text{twist}} \wedge \frac{J_{6\text{twist}}^2}{2!} \\
&= \int_{Y_7} \eta_{\text{twist}} \wedge b_1 \mathrm{d}\eta_{\text{twist}} \wedge \frac{(\omega_{\text{twist}} + A\mathrm{vol}_\Sigma)^2}{2!} \\
&= A \int_{Y_5} \eta \wedge \rho_6 \wedge \omega + 2\pi \sum_{i=1}^3 n_i \int_{Y_5} \eta \wedge \omega \wedge \left(x_i \rho_6 + b_1 w_i \omega\right),
\end{aligned}
\tag{5.13}
$$

where we have abused notation and denoted the forms $\eta$ and $\rho_6$ and their restrictions to $Y_5$ by the same symbol. To get the second equality we used the F-theory relation (5.12), which effectively reduces the expression to its constant axio-dilaton counterpart. The action is then identical to the case without the auxiliary elliptic fibration, apart from the fact that $n_1$, given in (5.9), depends on the degree of the duality line bundle. As mentioned above, the action can be rewritten in terms of derivatives of the master volume (5.3). Since we have shown that the F-theory action reduces to the constant axio-dilaton case, except for the dependence of $n_1$ on the duality line bundle, we can read off the result from [48]

$$S_F = -A \sum_{a=1}^d \frac{\partial \mathcal{V}}{\partial \lambda_a} - 2\pi b_1 \sum_{i=1}^3 n_i \frac{\partial \mathcal{V}}{\partial b_i}\,. \tag{5.14}$$

Under the twist the constraint equation becomes

$$
\begin{aligned}
0 &= \int_{Y_7} \eta_{\text{twist}} \wedge [\rho_6 - 2\pi c_1(\mathcal{L})]^2_{\text{twist}} \wedge J_{6\text{twist}} \\
&= A \int_{Y_5} \eta \wedge \rho_6^2 + \int_{Y_7} \eta_{\text{twist}} \wedge (b_1 \mathrm{d}\eta_{\text{twist}})^2 \wedge \omega_{\text{twist}} \\
&= A \int_{Y_5} \eta \wedge \rho_6^2 + 2\pi \sum_{i=1}^3 n_i \int_{Y_5} \eta \wedge \rho_6 \wedge (x_i \rho_6 + 4 b_1 w_i \omega) \,.
\end{aligned}
\tag{5.15}
$$

In terms of the master volume the equation is

$$
A \sum_{a,b=1}^d \frac{\partial^2 \mathcal{V}}{\partial \lambda_a \partial \lambda_b} - 2\pi n_1 \sum_{a=1}^d \frac{\partial \mathcal{V}}{\partial \lambda_a} + 2\pi b_1 \sum_{a=1}^d \sum_{i=1}^3 n_i \frac{\partial^2 \mathcal{V}}{\partial \lambda_a \partial b_i} = 0 \,.
\tag{5.16}
$$

Finally, we turn our attention to the flux integers. There are two types of five-cycles in $Y_7$. The first type are torus invariant three-cycles $S_a \subset Y_5$ fibered over $\Sigma$, which schematically will be written as $(S_a \to \Sigma)$. The second is $Y_5$ itself. The latter does not receive any contributions from the Riemann surface, since the curvatures $F_i$ and $c_1(\mathcal{L})$ integrate to zero on $Y_5$. We find

$$
\nu_3 N = \int_{Y_5} \eta \wedge \rho_6 \wedge \omega = -\sum_{a=1}^d \frac{\partial \mathcal{V}}{\partial \lambda_a} \,.
\tag{5.17}
$$

For the other class of five-cycles, the flux quantization conditions are given by

$$
\begin{aligned}
\nu_3 M_a &= \int_{(S_a \to \Sigma)} \eta_{\text{twist}} \wedge [\rho_6 - 2\pi c_1(\mathcal{L})]_{\text{twist}} \wedge J_{6\text{twist}} \\
&= \int_{(S_a \to \Sigma)} \eta_{\text{twist}} \wedge b_1 \mathrm{d}\eta_{\text{twist}} \wedge (\omega_{\text{twist}} + A \text{vol}_\Sigma) \\
&= A \int_{S_a} \eta \wedge \rho_6 + 2\pi \sum_{i=1}^3 n_i \int_{S_a} \eta \wedge (2 b_1 w_i \omega + x_i \rho_6) \\
&= \frac{A}{2\pi} \sum_{b=1}^d \frac{\partial^2 \mathcal{V}}{\partial \lambda_a \partial \lambda_b} + b_1 \sum_{i=1}^3 n_i \frac{\partial^2 \mathcal{V}}{\partial \lambda_a \partial b_i} \,.
\end{aligned}
\tag{5.18}
$$

The extremization procedure now amounts to explicitly solving the topological constraint (5.16) and flux quantization conditions (5.17) and (5.18) for the Kähler parameters $A, \lambda_a$ and R-symmetry vector $b_i$ and subsequently extremizing the action (5.14) with respect to the remaining free parameters.

## 5.3 M-Theory $\mathcal{I}$-Extremization for Toric Fibrations

In this section, we establish the $\mathcal{I}$-extremization procedure dual to the $c$-extremization for fibered toric geometries set up in the previous section. In the context of M-theory, we are

considering the physical compactification space

$$
\begin{array}{ccc}
 & & Y_9^\tau \\
 & \swarrow & \\
\mathbb{E}_\tau \hookrightarrow & B_4^\tau & \\
 & \sigma \big\uparrow \big\downarrow \pi & \\
 & \Sigma &
\end{array}
\qquad\qquad . \tag{5.19}
$$

Here we view $Y_9^\tau$ as the space $Y_5$ fibered over the elliptic surface $B_4^\tau$. This is achieved in the same way as in (5.4), but the vector bundle $\mathcal{O}(\vec{n})$ is now pulled back from $\Sigma$ to $B_4^\tau$. The cone $C(Y_9^\tau)$ also admits a non-vanishing holomorphic volume form precisely if

$$
n_1 = 2 - 2g - \deg \mathcal{L} . \tag{5.20}
$$

This geometric setup fits into the framework of [49]. In what follows we will make use of the formulas for toric fibrations over a general complex surface derived in that paper and specialize them to the elliptic surface case. We make the following ansatz for the Kähler form on $B_4^\tau$

$$
J_{B_4^\tau} = k_0 \omega_0 + \left( A + \frac{k_0 \deg \mathcal{L}}{2} \right) \mathrm{vol}_\Sigma . \tag{5.21}
$$

In other words, we are assuming that the Kähler class on the elliptic surface is just a linear combination of the base and the fiber class. One can also derive similar formulas for a more general ansatz where Cartan divisors are added. The choice of the shift of $A$ by $k_0 \deg \mathcal{L}/2$ is convenient in order to compare to the F-theory parameters at the end of this section. Using the ansatz the volume of the elliptic surface is

$$
\mathrm{vol}(B_4^\tau) = \int_{B_4^\tau} \frac{J_{B_4^\tau}^2}{2} = -\frac{k_0^2 \deg \mathcal{L}}{2} + \left( A k_0 + \frac{k_0^2 \deg \mathcal{L}}{2} \right) = A k_0 , \tag{5.22}
$$

where we used

$$
\int_{B_4^\tau} \omega_0^2 = -\deg \mathcal{L} , \qquad \int_{B_4^\tau} \omega_0 \wedge \mathrm{vol}_\Sigma = 1 . \tag{5.23}
$$

Furthermore, the curvature integrals specialize to

$$
\int_\Sigma F_i = 2\pi n_i , \qquad \int_{\mathbb{E}_\tau} F_i = 0 , \tag{5.24}
$$

and $F_i \wedge F_j = 0$ for dimensional reasons. With these results the M-theory constraint equation reduces to

$$
A \sum_{a,b=1}^d \frac{\partial^2 \mathcal{V}}{\partial \lambda_a \partial \lambda_b} - 2\pi n_1 \sum_{a=1}^d \frac{\partial \mathcal{V}}{\partial \lambda_a} + 2\pi b_1 \sum_{a=1}^d \sum_{i=1}^3 n_i \frac{\partial^2 \mathcal{V}}{\partial \lambda_a \partial b_i} = 0 , \tag{5.25}
$$

which is exactly the same as the F-theory constraint given in (5.16). The M-theory super-symmetric action becomes

$$S_M = -Ak_0 \sum_{a=1}^{d} \frac{\partial \mathcal{V}}{\partial \lambda_a} - 2\pi k_0 b_1 \sum_{i=1}^{3} n_i \frac{\partial \mathcal{V}}{\partial b_i} \,. \tag{5.26}$$

Let us now focus on the flux quantization conditions. The seven-cycles fall into two classes, where the cycles in the first class are obtained by fibering $Y_5$ over a two-cycle in the base, and the second class contains three-cycles in $Y_5$ (associated with toric divisors on the cone) fibered over the entire base. The flux integer corresponding to fixing a point in $\Sigma$ and quantizing over the cycle $Y_5 \times \mathbb{E}_\tau$ is

$$\nu_4 N = -k_0 \sum_{a=1}^{d} \frac{\partial \mathcal{V}}{\partial \lambda_a} \,. \tag{5.27}$$

The quantization conditions associated to fibrations of toric three-cycles over $B_4^\tau$ are

$$\nu_4 M_a = \frac{Ak_0}{2\pi} \sum_{b=1}^{d} \frac{\partial^2 \mathcal{V}}{\partial \lambda_a \partial \lambda_b} + k_0 b_1 \sum_{i=1}^{3} n_i \frac{\partial^2 \mathcal{V}}{\partial \lambda_a \partial b_i} \,. \tag{5.28}$$

There is one final cycle we need to consider, arising from the section of the elliptic fibration. Geometrically this is the space $Y_5$ fibered over $\Sigma$ and the corresponding flux number is given by

$$\nu_4 N_0 = -\left( A - \frac{k_0 \deg \mathcal{L}}{2} \right) \sum_{a=1}^{d} \frac{\partial \mathcal{V}}{\partial \lambda_a} - 2\pi b_1 \sum_{i=1}^{3} n_i \frac{\partial \mathcal{V}}{\partial b_i} \,. \tag{5.29}$$

Notice that, combining the expressions for the flux numbers, the supersymmetric action can be rewritten as

$$S_M = k_0 \nu_4 \left( N_0 + \frac{1}{2} N \deg \mathcal{L} \right) \,. \tag{5.30}$$

This toric setup provides an instructive example of the M/F-theory relations we have described in section 4. In the ansatz for the Kähler class (5.21) we have explicitly included the $k_0$ corrections to the Kähler class on the base of the elliptic fibration. Indeed, the parameter $A$ is precisely the F-theory Kähler parameter on $\Sigma$ and the relation

$$\lim_{k_0 \to 0} J_{B_4^\tau} = \lim_{k_0 \to 0} \left( k_0 \omega_0 + J_\Sigma + \frac{k_0 \deg \mathcal{L}}{2} \mathrm{vol}_\Sigma \right) = J_\Sigma \tag{5.31}$$

holds. The way to derive the explicit form of the correction term is to start with a general ansatz for the Kähler form on $B_4^\tau$ and impose that the $\mathcal{O}(k_0^2)$ terms in the M-theory constraint equation cancel. In this way the constraint equation reduces just to the linear term, which

is precisely the F-theory constraint equation. Moreover, the flux integers $M_a$ and $N$ also match on both sides if we take into account the relation $\nu_3 = \nu_4/2k_0$, as well as the fact that the master volume functions differ by a factor of 2. The detailed comparison of metrics and normalization in M- versus F-theory is discussed in appendix A.

An interesting feature of these geometries is that, despite including the full backreaction of the 7-branes in the M-theory background, there are no $k_0$ corrections in the supersymmetric action, i.e. we find

$$S_M = 2k_0 S_F \,. \tag{5.32}$$

This implies that the resulting on-shell solutions will have

$$\frac{1}{4G_2} = \frac{\Delta\phi}{12} c_{\text{sugra}} \tag{5.33}$$

on the nose, even though we are considering a non-trivial elliptic fibration. This is precisely the relation (4.14). We also see that this relation actually holds off-shell. The upshot of this discussion is that $\mathcal{I}$- and $c$-extremization are indeed equivalent for toric fibrations over a Riemann surface.

## 5.4 Universal Twist: Elliptic Surface

In this section, we focus on a known class of F-theory supergravity solutions found in [2], the so-called universal twist solution for elliptic surfaces. We apply the holographic $\mathcal{I}/c$-extremization developed in sections 2 and 3, which allows us to simultaneously re-derive the central charge of the 2d field theory dual to these F-theory solutions and determine $1/G_2$ of their M-theory duals, without ever explicitly solving the master equation.

The universal twist solutions are based on the ansatz

$$
\begin{array}{ccc}
S^1 & \lhook\joinrel\longrightarrow & Y_7 \\
& & \Big\downarrow \\
& & \Sigma \times \mathcal{M}_4
\end{array}
\tag{5.34}
$$

which assumes that the transverse Kähler space $\mathcal{M}_6$ is a product of a complex curve and a Kähler surface. We are interested in the set of universal twist solutions where the elliptic fibration is non-trivial only over the complex curve, so that $\mathcal{M}_8^\tau$ contains an elliptic surface

$$\mathcal{M}_8^\tau = (\mathbb{E}_\tau \to \Sigma) \times \mathcal{M}_4 \,. \tag{5.35}$$

This corresponds to choosing the twist parameters $n_i$ parallel to the R-symmetry vector, i.e. we take

$$n_i = \frac{n_1}{b_1} b_i \,, \tag{5.36}$$

which immediately implies

$$\sum_{a,b=1}^{d} \frac{\partial^2 \mathcal{V}}{\partial \lambda_a \partial \lambda_b} = 8 b_1^2 \text{Vol}(Y_5)\,, \qquad b_1 \sum_{i=1}^{3} n_i \frac{\partial \mathcal{V}}{\partial b_i} = -n_1 \mathcal{V}\,. \tag{5.37}$$

The topological constraint, which must be imposed for either side of the duality, is

$$8 A\, b_1^2 \, \text{Vol}(Y_5) - 4\pi n_1 \sum_{a=1}^{d} \frac{\partial \mathcal{V}}{\partial \lambda_a} = 0\,. \tag{5.38}$$

The M/F-theory flux integers are given by

$$\begin{aligned} \nu_3 N &= -\sum_{a=1}^{d} \frac{\partial \mathcal{V}}{\partial \lambda_a}\,, \\ \nu_3 M_a &= \frac{A}{2\pi} \sum_{b=1}^{d} \frac{\partial^2 \mathcal{V}}{\partial \lambda_a \partial \lambda_b} - n_1 \frac{\partial \mathcal{V}}{\partial \lambda_a}\,. \end{aligned} \tag{5.39}$$

Here we have chosen to write the quantization conditions manifestly as F-theory equations.[7] The distinguished M-theory flux integer is $N_0$, which satisfies

$$\nu_4 N_0 = -\left( A - \frac{k_0 \deg \mathcal{L}}{2} \right) \sum_{a=1}^{d} \frac{\partial \mathcal{V}}{\partial \lambda_a} + 2\pi n_1 \mathcal{V}\,. \tag{5.40}$$

The M/F-theory supersymmetric actions are

$$\begin{aligned} S_F &= A\,\nu_3 N + 2\pi n_1 \mathcal{V}\,, \\ S_M &= A\,\nu_4 N + 2\pi k_0 n_1 \mathcal{V}\,, \end{aligned} \tag{5.41}$$

so that again equation (5.32) holds, on the nose.

With these relations in place, we proceed to impose all common M/F-theory conditions (i.e. all but the $N_0$ flux quantization) and derive expressions for the supersymmetric actions that take these conditions into account. We start by rewriting the topological constraint in terms of the flux integer $N$ as

$$2 A\, b_1^2 \, \text{Vol}(Y_5) + \pi n_1 \nu_3 N = 0\,. \tag{5.42}$$

---

[7]We use the convention that whenever the parameter $\nu_4/\nu_3$ appears in an equation, the volumes $\mathcal{V}$ and $\text{Vol}(Y_5)$ are implicitly understood to be functions of $\vec{b}_M/\vec{b}_F$ and the equation itself should be understood as an M/F-theory equation. To write an equation as it appears naturally in the dual description, one simply uses $\nu_3 = \nu_4/2k_0$ and the normalization conventions detailed in appendix A. Equations where neither parameter appears are invariant under $\vec{b}_M \leftrightarrow \vec{b}_F$.

Solving this constraint for $A$ and substituting into the supersymmetric actions yields

$$S_F = -\frac{\pi n_1 \nu_3^2 N^2}{2b_1^2 \text{Vol}(Y_5)} + 2\pi n_1 \mathcal{V},$$

$$S_M = -\frac{\pi n_1 \nu_4^2 N^2}{2b_1^2 k_0 \text{Vol}(Y_5)} + 2\pi k_0 n_1 \mathcal{V}.$$

(5.43)

We can impose quantization of the $M_a$ by choosing $\lambda_a \equiv \lambda$. This ensures that the $M_a$ are quantized as

$$M_a = -N.$$

(5.44)

This solution implies that the master volume is

$$\mathcal{V} = 4b_1^2 \lambda^2 \text{Vol}(Y_5),$$

(5.45)

and the flux quantization condition for $N$ fixes $\lambda$ to be

$$\lambda = -\frac{\nu_3 N}{8b_1^2 \text{Vol}(Y_5)}.$$

(5.46)

The supersymmetric actions can then be written as

$$S_F = -\frac{3\pi n_1 \nu_3^2 N^2}{8b_1^2 \text{Vol}(Y_5)},$$

$$S_M = -\frac{3\pi n_1 \nu_4^2 N^2}{8k_0 b_1^2 \text{Vol}(Y_5)}.$$

(5.47)

Since $\text{Vol}(Y_5)$ is extremized for a Reeb vector with $r_1 = 3$, we set the M/F-theory R-symmetry vector $\vec{b}_F = \frac{2}{3}\vec{r} = 2\vec{b}_M$. Let $\vec{r}_*$ denote the extremal Reeb vector, corresponding to a Sasaki-Einstein metric on $Y_5$. We thus find the actions

$$S_F(\vec{r}_*) = -\frac{\pi n_1 \nu_3^2 N^2}{36 \text{Vol}(Y_5)(\vec{r}_*)},$$

$$S_M(\vec{r}_*) = -\frac{\pi n_1 \nu_4^2 N^2}{72 k_0 \text{Vol}(Y_5)(\vec{r}_*)}.$$

(5.48)

The 2d central charge is then given by (2.26) as

$$c_{\text{sugra}} = \frac{12(2\pi)^2}{\nu_3^2} S_F(\vec{r}_*) = -\frac{(2\pi)^3 n_1 N^2}{6 \text{Vol}(Y_5)(\vec{r}_*)}.$$

(5.49)

The AdS$_2$ Newton constant is given by (3.21) as

$$\frac{1}{G_2} = \frac{8(2\pi)^2}{\nu_4^{3/2}} S_M(\vec{r}_*) = -\frac{4\pi^3 n_1 \Delta\phi N^2}{9 \text{Vol}(Y_5)(\vec{r}_*)} = \Delta\phi \frac{c_{\text{sugra}}}{3},$$

(5.50)

as expected. However, this expression for the Newton constant cannot yet be understood to reflect a physical quantity due to the presence of $\Delta\phi$, which is a parameter of the internal space. In order for this to constitute a genuine M-theory solution, we still need to impose the $N_0$ flux quantization condition

$$N_0 = -\frac{\pi n_1 \nu_4 N^2}{72 k_0^2 \mathrm{Vol}(Y_5)(\vec{r}_*)} - \frac{1}{2} N \deg \mathcal{L} \,. \tag{5.51}$$

This condition fixes the period $\Delta\phi$ in terms of the distinguished flux number. We find

$$\Delta\phi = \pm \frac{6}{N} \sqrt{\frac{\mathrm{Vol}(Y_5)(\vec{r}_*) \left(2N_0 + N \deg \mathcal{L}\right)}{-\pi n_1}} \,. \tag{5.52}$$

The Newton constant of this genuine M-theory solution is then

$$\frac{1}{G_2} = \frac{8\pi^2 N}{3} \sqrt{\frac{-\pi n_1 \left(2N_0 + N \deg \mathcal{L}\right)}{\mathrm{Vol}(Y_5)(\vec{r}_*)}} \,. \tag{5.53}$$

## 5.5 Baryonic Twist: $Y^{p,q}$

We now consider the so-called baryonic twist solutions [2, 48]. For simplicity we present the computations for $Y_5 = Y^{p,q}$. The $Y^{p,q}$ metrics first appeared in [50] and their toric data was derived in [51]. The $d = 4$ ordered inward pointing normal vectors are

$$v_1 = (1,0,0), \quad v_2 = (1,1,0), \quad v_3 = (1,p,p), \quad v_4 = (1,p-q-1,p-q) \,. \tag{5.54}$$

The $Y^{p,q}$ metrics have $p > q > 0$ and the polyhedral cone with vectors $v_a$, $a = 1,\ldots,4$ is convex. We take the free twist parameters to be $n_2 = n_3 \equiv n$ for simplicity. As it turns out, the computational complexity of the problem is highly sensitive to the order in which the topological condition and flux quantization conditions are imposed, even though the resulting solution is clearly independent of this choice. We therefore include details of how the sets of equations are solved on each side of the duality.

We first discuss the F-theory side. We proceed by using (5.54) to explicitly write down an expression for the master volume $\mathcal{V}$, which is then a function of $\lambda_a$ and $b_i$. We then derive expressions for the constraint, fluxes and action in terms of $\mathcal{V}$ and set $b_1 = 2$. We use the flux quantization conditions for $N$ and $M_1$ to solve for $\lambda_4$ and $A$, respectively, and solve the constraint equation for $\lambda_1$. Note that $\lambda_2, \lambda_3$ must necessarily drop out of any final result, since there are only two independent Kähler parameters. We rescale the fluxes and twist parameters as

$$M_a \equiv -n_1 m_a N, \qquad n \equiv -n_1 s \,, \tag{5.55}$$

and immediately rename $m_1 \equiv m$ for notational convenience. The remaining fluxes are

$$m_2 = m_4 = \frac{(1-m)p+s}{p+q}, \qquad m_3 = \frac{(m-1)(p-q)-2s}{p+q} . \tag{5.56}$$

We can then determine the trial central charge, which we do not quote here as the expression is extremely long. Extremizing with respect to $b_2, b_3$ gives the R-symmetry vector $\vec{b} = (2, b_2, b_2)$ with

$$b_2 = -2p\frac{p^3\left[2(m-1)^2q+(2m-3)s\right]-2p^2\left[(m-1)^2q^2+(2m-1)qs+2s^2\right]+pqs[(2m-3)q-2s]-2q^2s^2}{p^4(2m-1)+2p^3[(2m-1)q+s]+p^2\left[(4m^2-6m+3)q^2+4mqs+4s^2\right]+2pqs[(3-2m)q+2s]+4q^2s^2} , \tag{5.57}$$

and on-shell central charge

$$c_{\text{sugra}} = \frac{12N^2n_1p[(m-1)p-s]\left[p^3\left(2m^2-3m+1\right)+p^2\left(\left(-2m^2+3m-1\right)q-4ms+s\right)+pqs(2m-3)-2qs^2\right]}{p^4(2m-1)+2p^3[(2m-1)q+s]+p^2\left[(4m^2-6m+3)q^2+4mqs+4s^2\right]+2pqs[(3-2m)q+2s]+4q^2s^2} . \tag{5.58}$$

Note that, for a trivial line bundle with $\deg \mathcal{L} = 0$, we find that $b_2$ and $c_{\text{sugra}}$ reduce to (6.6) and (6.7) in [48].

On the M-theory side we again start by explicitly writing down the master volume $\mathcal{V}$ as a function of $\lambda_a$ and $b_i$ using (5.54). We then derive expressions for the constraint, fluxes and action in which we set $b_1 = 1$. Noticing that the constraint equation does not depend on $\lambda_1$ and the flux quantization condition for $M_1$ does not depend on $\lambda_3$, we first solve the constraint equation for $\lambda_3$ and use the $M_1$ flux quantization condition to solve for $\lambda_1$. We then solve the $N$ flux quantization condition for $A$. Note that $\lambda_2, \lambda_4$ then automatically drop out of subsequent results, since there are only two independent Kähler parameters. Having imposed the topological constraint and flux quantization for $N$ and $M_1$, we reproduce the relations between the fluxes given in (5.56). The trial Newton constant can then be written down; however, the expression is not quoted here as it is very long. Extremizing with respect to $b_2, b_3$ gives the R-symmetry vector $\vec{b} = (1, b_2, b_2)$ with

$$b_2 = -p\frac{p^3\left[2(m-1)^2q+(2m-3)s\right]-2p^2\left[(m-1)^2q^2+(2m-1)qs+2s^2\right]+pqs[(2m-3)q-2s]-2q^2s^2}{p^4(2m-1)+2p^3[(2m-1)q+s]+p^2\left[(4m^2-6m+3)q^2+4mqs+4s^2\right]+2pqs[(3-2m)q+2s]+4q^2s^2} , \tag{5.59}$$

which is exactly half the corresponding R-symmetry component in F-theory, i.e. we have indeed found $\vec{b}_F = 2\vec{b}_M$. The preliminary Newton constant is

$$\frac{1}{G_2} = \frac{4\Delta_\phi N^2 n_1 p(-mp+p+s)\left[p^3\left(2m^2-3m+1\right)+p^2\left(\left(-2m^2+3m-1\right)q-4ms+s\right)+pqs(2m-3)-2qs^2\right]}{p^4(2m-1)+2p^3((2m-1)q+s)+p^2\left((4m^2-6m+3)q^2+4mqs+4s^2\right)+2pqs((3-2m)q+2s)+4q^2s^2} = \Delta_\phi\frac{c_{\text{sugra}}}{3} . \tag{5.60}$$

In order for this to correspond to a genuine M-theory solution, we must still impose quantization of $N_0$. Solving the flux quantization condition for $N_0$ for $k_0$, the Newton constant in terms of M-theory fluxes is

$$\frac{1}{G_2} = 8\pi N\sqrt{\frac{-pn_1[2N_0+\deg\mathcal{L}N][(m-1)p-s][p^3\left(2m^2-3m+1\right)+p^2\left(\left(-2m^2+3m-1\right)q-4ms+s\right)+pqs(2m-3)-2qs^2]}{p^4(2m-1)+2p^3((2m-1)q+s)+p^2\left((4m^2-6m+3)q^2+4mqs+4s^2\right)+2pqs((3-2m)q+2s)+4q^2s^2}} . \tag{5.61}$$

This concludes the discussion of solutions where $\mathcal{I}$- and $c$-extremization agree exactly across M/F-theory duality.

## 6  Universal Twist Solutions: Elliptic Three-fold

We would like to demonstrate that generically $1/G_2$ and $c_{\mathrm{sugra}}$ do not match exactly, as in the examples in section 5, but rather $1/G_2$ includes higher order corrections in $k_0$ as argued for in (4.11). These are absent in the F-theory solution, where the volume of the elliptic fiber is strictly zero. To this end, we consider the (on-shell) universal twist elliptic three-fold solutions, which were determined in [2]. We will first give a brief summary of the known F-theory solutions and then provide the corresponding M-theory analysis, and a comparison of the two.

### 6.1  F-Theory

The universal twist solutions are based on the product ansatz

$$
\begin{array}{ccc}
S^1 & \longrightarrow & Y_7 \\
& & \downarrow \\
& & \Sigma \times \mathcal{M}_4
\end{array}
\quad , \tag{6.1}
$$

where the transverse $\mathcal{M}_6$ factorizes as a product of a complex curve and a Kähler surface. To $\mathcal{M}_6$ we associate an auxiliary elliptic fibration $\mathcal{M}_8^\tau$, and assume that the fibration is non-trivial only over the $\mathcal{M}_4$ factor, so that the total space is given by

$$
\mathcal{M}_8^\tau = \Sigma \times (\mathbb{E}_\tau \to \mathcal{M}_4) \; . \tag{6.2}
$$

The metrics on $\Sigma$ and $\mathcal{M}_4$ satisfy

$$
\begin{aligned}
\rho_4 + \mathrm{d}Q &= 6 J_{\mathcal{M}_4} \, , \\
\rho_\Sigma &= -3 J_\Sigma \, .
\end{aligned}
\tag{6.3}
$$

Note that we assume that the Killing vector is regular throughout the following, and the period of the circle coordinate $z$ is $2\pi\ell$. The volume of $\Sigma$ is given by

$$
\mathrm{Vol}(\Sigma) = \frac{2\pi}{3}(2g - 2) \, , \tag{6.4}
$$

which follows from the Gauss-Bonnet theorem. Moreover,

$$
[J_{\mathcal{M}_4}] = \frac{\pi}{3} \left( c_1(\mathcal{M}_4) - c_1(\mathcal{L}) \right) \; . \tag{6.5}
$$

This equation implies that the volume of the circle-fibration over $\mathcal{M}_4$, denoted by $\mathcal{M}_5$, is

$$\mathrm{Vol}(\mathcal{M}_5) = \frac{\pi^3 \ell}{27} \int_{\mathcal{M}_4} (c_1(\mathcal{M}_4) - c_1(\mathcal{L}))^2 \,. \tag{6.6}$$

There are two classes of flux quantization conditions. The first corresponds to the cycle at fixed coordinates in $\Sigma$, which is a copy of $\mathcal{M}_5$. The flux integer is

$$\nu_3 N^F = 18\mathrm{Vol}(\mathcal{M}_5) \,. \tag{6.7}$$

The second class of flux quantization conditions is obtained as a $U(1)$ fibration over the product of $\Sigma$ with two-cycles $C_\alpha$ in $\mathcal{M}_4$. They are given by

$$\nu_3 M_\alpha^F = 3\pi \ell \mathrm{Vol}(\Sigma) C_\alpha \cdot [J_{\mathcal{M}_4}] \,. \tag{6.8}$$

Finally, the central charge of the 2d $(0,2)$ theory was computed in [2] to be

$$c_{\mathrm{sugra}} = \frac{2\pi^2 \mathrm{Vol}(\Sigma)(N^F)^2}{\mathrm{Vol}(\mathcal{M}_5)} \,. \tag{6.9}$$

## 6.2   M-Theory

We start with the F-theory metric on $\mathcal{M}_6$ and construct the M-theory solution from it. We will specialize to the case where $\mathcal{M}_4 = \mathbb{CP}^2$. Consider the Kähler class ansatz

$$J_8 = k_0 \omega_0 + x J_\Sigma + y J_{\mathcal{M}_4}. \tag{6.10}$$

The cruical point is that the form of the metrics on $\Sigma$ and $\mathcal{M}_4$ are exactly the same as in the previous subsection, and we think of the F-theory solution as a 0-th order solution in a suitable expansion in the volume of the elliptic fiber. We have introduced $x$ and $y$ that parametrize the Kähler cone of $\mathcal{M}_6 = \Sigma \times \mathcal{M}_4$. Given this parametrization we now compute the Kähler class of the M-theory solution.

Note that the Kähler class on $\mathcal{M}_4$ is

$$[J_{\mathcal{M}_4}] = \frac{\pi}{3}(3 - \deg \mathcal{L})[H] \,, \tag{6.11}$$

where $[H]$ is the hyperplane class of $\mathbb{CP}^2$. The line bundle associated to the elliptic fibration lives over $\mathcal{M}_4$ and in particular $c_1(\mathcal{L})^2 \neq 0$. From the M-theory constraint we derive the following equation

$$x = y - \frac{3k_0 \deg \mathcal{L}}{2\pi(3 - \deg \mathcal{L})} \,, \tag{6.12}$$

where

$$\deg \mathcal{L} = c_1(\mathcal{L}) \cdot [H] \,. \tag{6.13}$$

This allows us to eliminate the parameter $x$ from the above ansatz. With this we can then compute the M-theory supersymmetric action

$$S_M = \frac{4\pi^2 \ell (g-1)}{3} k_0 \left[ \pi^2 y^2 (3 - \deg \mathcal{L})^2 - 3\pi y (3 - \deg \mathcal{L}) \deg \mathcal{L} k_0 + 2 \deg \mathcal{L}^2 k_0^2 \right] . \quad (6.14)$$

The remaining parameters, $y$ and $k_0$ are fixed by flux quantization in M-theory

$$\nu_4 N_0 = \frac{4\pi^2 \ell (g-1)}{3} \left[ \pi^2 y^2 (3 - \deg \mathcal{L})^2 - 4\pi y (3 - \deg \mathcal{L}) \deg \mathcal{L} k_0 + 3 \deg \mathcal{L}^2 k_0^2 \right] ,$$

$$\nu_4 N^M = \frac{4\pi^3 \ell}{3} y k_0 (3 - \deg \mathcal{L})^2 + 2\pi^2 \ell (3 - \deg \mathcal{L}) \deg \mathcal{L} k_0^2 , \quad (6.15)$$

$$\nu_4 M^M = \frac{4\pi^2 \ell (g-1)}{3} \left[ 2\pi (3 - \deg \mathcal{L}) y k_0 + 3 \left( 1 - \frac{6}{(3 - \deg \mathcal{L})\pi} \right) \deg \mathcal{L} k_0^2 \right] .$$

Imposing the flux quantization condition for $N^M$ gives

$$y = \frac{3 \nu_4 N^M}{4\pi^3 \ell (3 - \deg \mathcal{L})^2 k_0} - \frac{3 \deg \mathcal{L}}{2\pi (3 - \deg \mathcal{L})} k_0 . \quad (6.16)$$

Let us briefly digress and examine this expression in more detail. Note that we can substitute $N^F$ into the above to obtain

$$y = \frac{N^M}{N^F} - \frac{3 \deg \mathcal{L}}{2\pi (3 - \deg \mathcal{L})} k_0 . \quad (6.17)$$

From this expression it is apparent that

$$\lim_{k_0 \to 0} J_8 = J_\Sigma + J_{\mathcal{M}_4} \quad (6.18)$$

holds if $N^M = N^F \equiv N$ are identified, as expected. Substituting (6.16) into the supersymmetric action results in

$$S_M = \frac{3(g-1)\nu_4^2 N^2}{4\pi^2 \ell (3 - \deg \mathcal{L})^2 k_0} - \frac{6\nu_4 N (g-1) \deg \mathcal{L}}{3 - \deg \mathcal{L}} k_0 + \frac{35\pi^2 \ell (g-1)}{3} (\deg \mathcal{L})^2 k_0^3 . \quad (6.19)$$

Using (3.21), we determine the leading contribution to the preliminary Newton constant to be

$$\frac{1}{G_2} = \frac{24(g-1)\sqrt{\nu_4} N^2}{\ell (3 - \deg \mathcal{L})^2 k_0} + \mathcal{O}(k_0) . \quad (6.20)$$

Comparing this with the expression for $c_{\text{sugra}}$ derived in the previous subsection, we indeed find

$$\frac{1}{G_2} = \frac{\Delta \phi}{3} c_{\text{sugra}} + \mathcal{O}(k_0) . \quad (6.21)$$

The distinguished flux number $N_0$ is given by

$$N_0 = \frac{3(g-1)\nu_4}{\pi^2 \ell (3 - \deg \mathcal{L})^2 k_0^2} N^2 - \frac{7(g-1) \deg \mathcal{L}}{3 - \deg \mathcal{L}} N + 15\pi^2 \ell (g-1) (\deg \mathcal{L})^2 \frac{k_0^2}{\nu_4} . \quad (6.22)$$

Substituting in the duality relations we find

$$N_0 = \left(\frac{\Delta\phi}{2\pi}\right)^2 \frac{c_{\text{sugra}}}{24} - \frac{7(g-1)\deg\mathcal{L}}{3-\deg\mathcal{L}}N + \frac{15\ell(g-1)(\deg\mathcal{L})^2}{4}\left(\frac{2\pi}{\Delta\phi}\right)^2. \qquad (6.23)$$

In particular, this is an expansion in $\Delta\phi$ where the quadratic term is proportional to $c_{\text{sugra}}$, again as expected. Finally, we record that

$$\begin{aligned}
\frac{1}{G_2} = \frac{16\pi^2}{81\deg\mathcal{L}} &\left[16\pi(3-\deg\mathcal{L})N_0 + 3\text{Vol}(\Sigma)N\deg\mathcal{L} + \mathbf{V}\right] \\
&\times \sqrt{\frac{2(3-\deg\mathcal{L})N_0 - 21\text{Vol}(\Sigma)N\deg\mathcal{L} + 2\mathbf{V}}{15(3-\deg\mathcal{L})\text{Vol}(\mathcal{M}_5)\text{Vol}(\Sigma)}},
\end{aligned} \qquad (6.24)$$

where

$$\mathbf{V} = \sqrt{4\pi^2(3-\deg\mathcal{L})^2N_0^2 + 42\pi\text{Vol}(\Sigma)N_0 N(3-\deg\mathcal{L})\deg\mathcal{L} + 9\left[N\text{Vol}(\Sigma)\deg\mathcal{L}\right]^2}. \qquad (6.25)$$

As a check on this expression we may formally expand it around the trivial torus fibration with $\deg\mathcal{L} = 0$

$$\frac{1}{G_2} = 8\pi^2 N\sqrt{\frac{N_0\text{Vol}(\Sigma)}{3\text{Vol}(\mathcal{M}_5)}} - 5\pi N^2\sqrt{\frac{\text{Vol}(\Sigma)^3}{3N_0\text{Vol}(\mathcal{M}_5)}}\deg\mathcal{L} + \mathcal{O}(\deg\mathcal{L}^2). \qquad (6.26)$$

The first term should then match (5.53), and using the expression (6.4) for $\text{Vol}(\Sigma)$ one can see that this is indeed the case.

## 7 Conclusions and Outlook

There has been much progress recently in tests of holography for 2d and 1d SCFTs. With decreasing number of supercharges on either side of the correspondence, the duality becomes more interesting and harder to study. The class of theories we discussed in this paper have the minimal amount of supersymmetry, whilst keeping a non-trivial R-symmetry. The $U(1)$ R-symmetry in 2d and 1d can mix with global $U(1)$ symmetries, and only after applying $c$- or $\mathcal{I}$-extremization is the true superconformal R-symmetry determined. The main paradigm in this paper was to study this problem holographically in the context of Type IIB solutions, where the axio-dilaton has a non-trivial spacetime-dependent profile – i.e. F-theory. We showed that the $c$-extremization of 2d SCFTs obtained from wrapped D3-branes in F-theory compactifications define a geometric extremization problem in the holographically dual AdS$_3$ solutions. This allowed us to compute, using an off-shell approach, the central charge of the SCFTs from holography.

As a counterpoint to the 2d SCFTs, we discussed 1d SCQMs obtained by M2-branes wrapped on complex curves, and the dual holographic $\mathcal{I}$-extremization principle in M-theory. By M/F-duality, whereby an elliptic fiber in the M-theory geometry becomes the auxiliary elliptic fibration of F-theory, these two setups can be related. The F-theory result for the central charge is obtained by considering the limit in M-theory where the volume of the elliptic fiber is taken to zero ($k_0 \to 0$). As we showed, there are classes of SCFTs where the resulting identification (1.1) is true without any higher order corrections in $k_0$ – these were discussed in section 5. In contrast, the class of solutions in section 6 showed that in general there can indeed be corrections to the F-theory expression of the central charge, in order for this to match the 1d partition function. Whenever both sides agree, the solution has an elliptic fibration that is non-trivial only over a complex curve. We observed that for elliptic fibrations over higher-dimensional base manifolds, there are generically correction terms, which arise from non-trivial higher intersection numbers on the base, e.g. from $c_1(\mathcal{L})^2$. This was exemplified in the elliptic three-folds of section 6.

Our analysis was largely focused on the geometric side of holography. Much is known about the wrapped D3-brane theories in F-theory, in terms of central charge computations. However much less understood is the precise relation between the dimensional reduction of such 2d SCFTs with the 1d SCQM that arises from the dual M2-brane configuration, and the associated 1d partition function. Related computations are known for higher (and non-chiral) supersymmetric theories, but for $(0, 2)$ this remains an exciting open problem.

### Acknowledgements

We thank C. Closset, C. Couzens, J. Eckhard, H. Kim for discussions. SC and SSN are supported by the ERC Consolidator Grant number 682608 "Higgs bundles: Supersymmetric Gauge Theories and Geometry (HIGGSBNDL)".

## A    Comparison of Normalizations in M/F-Theory

To streamline the notation in the main text, we implicitly always assume a particular normalization in M-theory and in F-theory. The purpose of this appendix is to explain these normalizations. We uniformly denote the compact spaces by $Y_7$ and $Y_9^\tau$ throughout the paper, despite the fact that the metrics on these spaces differ depending on whether they appear in F- or M-theory: the normalization of the Killing vectors $\xi$ differ by a factor of 2

$$\text{F-theory/IIB} : \xi = 2\partial_z \,, \qquad \text{M-theory} : \xi = \partial_z \,. \tag{A.1}$$

Therefore, the $b_i$ coefficients in the parametrization in (2.44) are related by $\vec{b}_F = 2\vec{b}_M$. The Killing one-form $\eta$ is correspondingly normalized as

$$\text{F-theory/IIB}: \eta = \frac{1}{2}(\mathrm{d}z + P), \qquad \text{M-theory}: \eta = \mathrm{d}z + P. \tag{A.2}$$

The metrics on the compact spaces are given by

$$\mathrm{d}s^2(Y_7) = \eta^2 + \mathrm{e}^B \mathrm{d}s^2(\mathcal{M}_6), \qquad \mathrm{d}s^2(Y_9^\tau) = \eta^2 + \mathrm{e}^B \mathrm{d}s^2(\mathcal{M}_8^\tau), \tag{A.3}$$

with the warp factor and normalization of $\eta$ pertaining to F- or M-theory. Furthermore, we have also used the same symbol $F$ for the closed two-form appearing in the fluxes $F_5$ in (2.1) and $G_4$ in (3.14) in F- and M-theory, respectively. It is given by

$$\text{F-theory/IIB}: F = -2J_6 + \mathrm{d}\left(e^{-B_{10}}\eta\right), \qquad \text{M-theory}: F = -J_8 + \mathrm{d}\left(e^{-B_{11}}\eta\right), \tag{A.4}$$

with the warp factor, normalization of $\eta$ and Kähler form pertaining to F- or M-theory. Finally, in the toric examples of section 5, the volumes $\mathcal{V}$ and $\mathrm{Vol}(Y_5)$ implicitly depend on the R-symmetry vector $\vec{b}$ so that

$$\mathcal{V}(\vec{b}_M) = 2\mathcal{V}(\vec{b}_F), \qquad \mathrm{Vol}(Y_5)(\vec{b}_M) = 2^3 \mathrm{Vol}(Y_5)(\vec{b}_F). \tag{A.5}$$

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
