# Peer review of "$\mathcal{I}$/$c$-Extremization in M/F-Duality"

_SciPost Physics_

## Round 2 · Referee Report · Anonymous (Referee 1) · 2020-6-23

Report

This is an interesting paper where the geometric counterpart of c-extremization is compared with the geometric counterpart of I-extremization for a class of dual F-theory and M theory backgrounds. The paper contains many interesting technical results and examples that certainly deserve to be published. Before publication I would like to ask the author to address a point and clarify a confusion I have.

The authors start with an AdS3 background in F-theory, T-dualize along a circle inside AdS3 and obtain an AdS2 background in M theory. I expect this background to be the horizon geometry of black hole. And indeed I-extremization has been introduced in this context. In particular, some of the examples in section 5, after reduction on the Sasaki-Einstein manifold, can be probably thought of as black strings in 5d with horizon AdS3 times a Riemann surface.
Now in this context there is a standard way of going from black strings (AdS3) to a black hole (adS2): compactify on a circle in AdS3 and add a momentum n along it. The entropy of the black hole is then related to the CFT central charge by the Cardy formula: S=Sqrt{n c}, which is obtained as the
Legendre transform of the logarithm of the partition function of the CFT:
S= log Z + beta n = c/beta + n beta -> n=c/beta^2 and S ~ Sqrt(n c) (*)
where beta ~ tau is the modular parameter of the torus.

How this construction is related with what the authors do? In particular, the author have an integer N0 on the AdS2 side with no counterpart in F theory, which is a bit strange. Can N0 be related to a construction similar to the above mentioned one? In particular, I notice analogies with (*) in the right-hand side of (1.1) and in (4.9) if we identify Delta phi=1/beta.
  • validity: -
  • significance: -
  • originality: -
  • clarity: -
  • formatting: -
  • grammar: -

Author:  Sebastjan Cizel  on 2020-07-01  [id 868]

(in reply to Report 1 on 2020-06-23)
Category:
remark
answer to question

We thank the referee for their report on our paper.

The referee makes an interesting observation, that perhaps there is an alternative interpretation of the flux parameter $N_0$ in terms of a momentum along a circle compactification of a black string. Although the expressions in (1.1) and (4.9) might suggest this, we are not convinced that this is a general interpretation of the parameter $N_0$. First of all, our analysis is not only restricted to black string solutions but, more importantly, as we explain in the paper, the parameter $N_0$ is a flux on the M-theory side, which to our knowledge does not (in general) have an interpretation as a momentum.

The referee seems to indicate that we are lacking an interpretation of the parameter $N_0$ on the F-theory side. We should emphasize that this is not the case: As explained around (4.9), and more explicitly in later examples, the flux quantization of $N_0$ fixes the period $\Delta \phi$ of the circle in F-theory. In this sense this is a complete dictionary, where there is no remaining mystery in terms of the supergravity analysis. We hope this clears up this confusion.

Anonymous on 2020-07-05  [id 873]

(in reply to Sebastjan Cizel on 2020-07-01 [id 868])

The interpretation as a momentum would be on the F-theory side (essentially AdS3 replaced by BTZ) not on the M-theory one, but I understand that the black strings are only a subclass of solutions and I'm (partially) satisfied by the comment. I'm satisfied by the paper, in my opinion it can be published. I don't need to see it again.

---

## Editorial Decision

resubmitted